# GAN-GENERATED SAMPLE DETECTORS ARE STRONG PRIVACY ATTACKERS

## ABSTRACT

Since their inception Generative Adversarial Networks (GANs) have been popular generative models across images, audio, video, and tabular data. One promising application of generative models like GANs is to share restricted or sensitive data with third parties through the creation of synthetic data or by sharing the model itself. However, recent research on other types of generative models Carlini et al. (2023) has highlighted privacy vulnerabilities in this approach – namely that the models memorize significant quantities of the training data, and that existing membership inference attacks can identify generated samples as training points. This paper investigates the privacy implications of using GANs in black-box settings, where adversaries only have access to samples from the generator, rather than access to the discriminator as is often assumed in prior work. We introduce a suite of membership inference attacks against GANs in the black-box setting and evaluate our attacks on image GANs trained on the CIFAR10 dataset and tabular GANs trained on genomic data. Our most successful attack, called The Detector, involve training a second network to score samples based on their likelihood of being generated by the GAN as opposed to a sample from the distribution. We prove under a simple model of the generator why detectors can be approximately optimal membership inference attacks. Across a wide range of tabular and image datasets, attacks, and GAN architectures we find that adversaries can orchestrate non-trivial privacy attacks when provided with access to samples from the generator. However, the observed privacy leakage in GANs appears to be lower compared to other generative and discriminative models.

## 1 INTRODUCTION

Beginning in 2014 with the seminal paper Goodfellow et al. (2014), until the recent emergence of diffusion models Sohl-Dickstein et al. (2015); Ho et al. (2020); Kong et al. (2020b); Ho et al. (2022), generative adversarial networks (GANs) have been the dominant generative model across image, audio, video, and tabular data generation Wang et al. (2020); Kong et al. (2020a); Liu et al. (2021); Wang et al. (2018); Xu et al. (2019b); Zhao et al. (2021), and are still in some cases competitive with diffusion models, for example in Kang et al. (2023b); Karras et al. (2021); Crowson et al. (2022); Hudson & Zitnick (2021). One promising use case of generative models is when the underlying training data is private, and so there are restrictions on sharing the data with interested parties. Rather than sharing the data directly, after training a generative model $G$ on the private data, either $G$ can be shared directly, or a new synthetic dataset can be generated from $G$, which can be used for downstream tasks without sharing the underlying data. While this seems private on the surface, a line of work has shown that sharing access to a generative model or synthetic dataset may leak private information from the underlying training data Chen et al. (2020); Hayes et al. (2018). Recent work Carlini et al. (2023) has shown that as compared to GANs, diffusion models are far less private, memorizing as much as $5\times$ more of their training data, and as a result are highly susceptible to attacks that can reconstruct the underlying training data. These considerations make GANs a more natural choice of generative model for highly sensitive data – or do they?

Most existing work probing the privacy properties of GANs has typically focused on the "white-box" setting, where an adversary has access to the discriminator of the GAN, and is able to use standard techniques to conduct membership inference attacks based on the loss Chen et al. (2020); a notable exception is van Breugel et al. (2023a). While these white-box attacks demonstrate releasing the

discriminator in particular leaks information about the underlying training set, they are unsatisfying from a practical perspective, since typically only the generator would be shared after training. This raises a natural question about the privacy of GANs: In the black-box setting where an adversary only has sample access to the generator, are effective privacy attacks possible?

Another potential motivation for developing efficient methods to determine whether a point was used to train a GAN comes from copyright. Suppose the creator of an image for example, suspects that their image was used in the training set of a generative model without their permission. Since the vast majority of state-of-the-art generative models are only made available via access to the generator, applying a black-box membership inference attack is the only way the creator can argue it is (statistically) likely their image was part of the training set. We address this gap by developing a battery of membership inference attacks against GANs that only require access to generated samples. We evaluate our attacks on target GANs trained on image and tabular data, focusing on image GANs trained on the popular CIFAR10 dataset Krizhevsky (2009), and on tabular GANs trained on SNP (Single Nucleotide Polymorphisms Gray et al. (2000)) data from two popular genomic databases Tryka et al. (2014); Consortium et al. (2015). We selected genomic data because of the obvious underlying privacy concerns, and because of the high-dimensionality of the data, which makes the setting more challenging from a privacy perspective. Early privacy attacks on genomic databases Homer et al. (2008a) led the NIH to take the extreme step of restricting access to raw genomic summary results computed on SNP datasets like dbGaP Tryka et al. (2014), although they later reversed course following scientific outcry.

**Contributions.** In this work, we develop state-of-the-art privacy attacks against GANs under the realistic assumption of black-box access, and conduct a thorough empirical investigation across 2 GAN architectures trained on 6 different tabular datasets created by sub-sampling 2 genomic data repositories, and 4 GAN architectures trained to generate images from CIFAR-10. Our main attack, which we call the Detector (and its extended version called the Augmented Detector or ADIS ), trains a second network to detect GAN-generated samples, and applies it to the task of membership inference. We derive novel theoretical results (Theorem 4.1) that shed light on our attack performance. We compare the Detector-based methods to an array of distance and likelihood-based attacks proposed in prior work Chen et al. (2020); Hayes et al. (2019); van Breugel et al. (2023b) which we are the first to evaluate with recent best practice metrics for MIAs Carlini et al. (2021). Generally, we find that while privacy leakage from GANs seems to be lower than reported in prior work Hayes et al. (2019); Carlini et al. (2023) for other types of generative models, there does still appear to be significant privacy leakage even in the black-box setting. Moreover, this privacy-leakage as measured by the success of our MIAs seems to vary based on the type of GAN trained, as well as the dimension and type of the underlying training data. In Section H we show how even when the adversary does not have access to exact reference samples from the distribution, Detector-based attacks are still effective given access to a different public genomic database. In Section I we explore differentially private (DP) GAN training in order to mitigate MIA attack success.

## 2 RELATED WORK

Membership Inference Attacks or MIAs were defined by Homer et al. (2008a) for genomic data, and formulated by Shokri et al. (2017) against ML models. In membership inference, an adversary uses model access as well as some outside information to determine whether a candidate point $x$ is a member of the model training set. Standard MIAs typically exploit the intuition that a point $x$ is more likely to be a training point if the loss of the model evaluated at $x$ is small Yeom et al. (2018), or in the white-box setting where they have access to the model's gradients, if the gradients are small Nasr et al. (2019), although other approaches that rely on a notion of self-influence Cohen & Giryes (2022) and distance to the model boundary have been proposed Pawelczyk et al. (2022); Choquette-Choo et al. (2020). The state of the art attack against discriminative classifiers, called LiRA Carlini et al. (2021), approximates the likelihood ratio of the loss when $x$ is in the training set versus not in the training set, training additional shadow models to sample from these distributions.

The most closely related work to this paper is in Hayes et al. (2019). First they show that loss-based attacks are easily ported over to GANs when the adversary has access to the discriminator loss. However, given that the discriminator is not required to sample from the generator once training is finished, there is no practical reason to assume that the adversary has access to the discriminator.

Hayes et al. (2019) find that by thresholding on the discriminator loss, TPR at FPR = .2 on DCGAN trained on CIFAR-10 is a little under .4 when the target GAN is trained for 100 epochs, but goes up to almost $100\%$ after 200 epochs, indicating severe over-fitting by the discriminator. In their Section 4.4 Hayes et al. (2019) also develop a black-box attack that is very similar to the Detector, where they train a discriminative model to classify samples from the GAN and test samples. While they conclude that this attack fails to outperform the random baseline, we reach a different conclusion by (i) evaluating on new genomic data settings (ii) combining this approach with distance-based attacks as part of our ADIS pipeline and (iii) doing a more fine-grained evaluation of MIA attack success by plotting $\log - \log$ ROC curves. Our empirical results show this discriminative approach significantly outperforms the random baseline, particularly at low FPRs on genomic data. There have been a handful of recent papers that propose what we call distance-based black-box attacks against generative models, which rely on the heuristic that if a point is "closer" to generated samples that to random points from the distribution, it is more likely to be a training point Hilprecht et al. (2019)Chen et al. (2020). van Breugel et al. (2023b) propose a density-based model called DOMIAS, which infers membership by targeting local over-fitting of the generative models. Rather than train a detector network to classify whether samples are generated from the target GAN or the reference data, DOMIAS performs dimension reduction in order to directly estimate both densities, and then uses the ratio of the densities as a statistic for membership inference. We implement many of these methods in our experiments section, and summarize them in more detail in Appendix J. Though less directly related to the MIAs developed in this paper, there is a flurry of recent work on balancing privacy during GAN training with utility Xie et al. (2018); Jordon et al. (2018); Long et al. (2021); Tantipongpipat et al. (2019); Rosenblatt et al. (2020); Liu et al. (2020); Mukherjee et al. (2020).

## 3 PRELIMINARIES

**Generative Adversarial Network(GANs).** Given training data drawn from a distribution, $X_T \sim \mathcal{P}$, a generative model tries to approximate $\mathcal{P}$. Generative adversarial networks (GANs) are examples of generative models and have gained popularity for their ability to generate realistic samples across a range of domains Arjovsky et al. (2017); Yu et al. (2020); Brock et al. (2018); Kang & Park (2021); Radford et al. (2016). The basic GAN set-up consists of two players - the discriminator ($D'$) and the generator ($\mathcal{G}$)- engaging in a minimax game Goodfellow et al. (2020) with training objective given as:

$$\mathbb{E}_{\mathbf{x} \sim \mathcal{P}}[\log D'(\mathbf{x})] + \mathbb{E}_{\mathbf{z} \sim p_{\mathbf{z}}(\mathbf{z})}[\log(1 - D'(G(\mathbf{z})))]$$

The generator, $G$, takes latent noise, $z$, as an input and generates a sample as output, while the discriminator examines data samples (i.e, $\mathbf{x}$, and $G(\mathbf{z})$) and tries to discriminate samples from the generator (i.e $G(\mathbf{z})$ ) from the real data samples (i.e $\mathbf{x}$). The generator's target is to generate realistic samples that would fool the discriminator, while the discriminator tries to detect counterfeit samples from the generator Goodfellow et al. (2020). The training objective is typically optimized by updating the generator and discriminator via simultaneous gradient ascent.

Throughout the paper we'll let $\mathcal{T}$ be the distribution that samples a random point from $X_T$, $\mathcal{G}$ the distribution that samples a point from $G$, $\mathcal{M} = \frac{1}{2}\mathcal{G} + \frac{1}{2}\mathcal{P}$ be the mixture distribution of $\mathcal{G}$ and $\mathcal{P}$, which we will need to define our detector attack in Section 4.1, and $\mathcal{R} = \frac{1}{2}\mathcal{T} + \frac{1}{2}\mathcal{P}$ be the mixture distribution we evaluate our MIAs on. Given a point $x$ let $\mathcal{G}(x), \mathcal{P}(x)$ denote the respective probability densities evaluated at $x$.

**Black-Box Setting.** Following convention, we assume that our black-box attacks have access to samples from $\mathcal{G}$ but not the weights of $G$ directly, access to details about how the GAN was trained (e.g. the architecture of course not the training samples), and fresh samples from the distribution $\mathcal{P}$ Shokri et al. (2017); Jia et al. (2019); Sablayrolles et al. (2019).

**Attack Framework.** We adopt an attack framework similar to that used in countless papers on membership inference attacks against machine learning models Shokri et al. (2017); Ye et al. (2021); Carlini et al. (2021). The model developer samples a training set $X_T \sim \mathcal{P}$, and trains a GAN $G$ on $X_T$, $G \sim \text{Train}(X_T)$. Now consider a membership inference attack (MIA) $\mathcal{A}$; draw $x' \sim \mathcal{R}$ and send $x'$ to $\mathcal{A}$. The membership inference attacker $\mathcal{A}$ receives the candidate point $x'$ and then outputs a guess: 1 if $x' \in X_T$, 0 if $x' \notin X_T$. Note that if $\mathcal{A}$ only receives $x'$, then it is impossible for $\mathcal{A}$ to output a guess that has accuracy $> \frac{1}{2}$, since the marginal distribution on $x'$ is always $\mathcal{P}$. Absent any

additional information, for an fixed FPR $\alpha$, the TPR achieved by $\mathcal{A}$'s guess is always $\leq \alpha$. In the black-box setting $\mathcal{A}$ also receives black-box access to draw random samples from $\mathcal{G}$ and additional samples from $\mathcal{P}$. From the above discussion, any accuracy $\mathcal{A}$ is able to achieve that is better than a random guess must be due to the ability to infer if $x' \in X_T$ through access to samples from $\mathcal{G}$. In practice, rather than an MIA $\mathcal{A}$ outputting a 1 or 0 guess, $\mathcal{A}$ will output a real-valued score $\mathcal{A}(x')$ corresponding to how likely $x'$ is to be in $X_T$. Then any threshold $\tau \in \mathbb{R}$ corresponds to an attack $\mathbf{1}\{\mathcal{A}(x') > \tau\}$, and varying $\tau$ traces out a receiver operating characteristic (ROC) curve of achievable FPR/TPRs Carlini et al. (2021).

**Attack Evaluation Metrics.**   Recent work Carlini et al. (2021); Liu et al. (2022) argues the most meaningful way to evaluate a membership inference attack $\mathcal{A}$ is not to look at the overall accuracy, but to focus on what TPRs are achieveable at low FPRs, as this could correspond to a more realistic privacy violation: a small subset of the training data that model can identify with high confidence. Note that this correspond to large values of $\frac{\text{TPR}}{\text{FPR}}$ at small FPRs. We adopt this convention here, reporting the achievable TPR at fixed FPRs (.001, .005, .01, .1) for all of our attacks (Tables 3, 6). As in Carlini et al. (2021) we plot all our attack ROC curves on the log-log scale in order to visualize what is happening at low FPRs. We also report the AUCs for each ROC curve in order to provide a quick high-level summary of attack success, but again we regard such average summary statistics as less important than the $\frac{\text{TPR}}{\text{FPR}}$ statistics.

## 4   ATTACK MODEL TYPES

The attacks we study fall into two broad categories: (i) Detector-based attacks rely on training a second classifier to distinguish generator samples from reference data samples, and then using that classifier's predicted probability of being generated by $\mathcal{G}$ as a proxy for training set membership (ii) Distance-based attacks compute the distance between the candidate point and samples from the generator and samples from the reference data, and predict the point is a training point if it is closer to generated samples. In Section 4.1 we outline the basic process of training the Detector, and define its augmented variant ADIS. We state theoretical results that characterize the performance of an optimal detector-based attack and show that under a simplified model where the generator learns the distribution subject to some mode collapse, detector-based attacks are approximately optimal MIAs. In Section 4.2 we define the distance-based attacks proposed in prior work that we evaluate our methods against. In our experiments we also compare against the DOMIAS attack proposed in van Breugel et al. (2023b) – we defer the details of our implementation to Appendix D.

### 4.1   DETECTOR-BASED ATTACKS

The Detector is based on the premise that a network that can distinguish samples generated from the GAN from samples from the true distribution, can also distinguish training samples from the distribution. Specifically we create a dataset by sampling GAN samples from $\mathcal{G}$ and labeling them 1, and generating samples from $\mathcal{P}$ and labeling them as 0. Then we train a detector network $D_\theta$ to classify GAN samples as 1 and samples from $\mathcal{P}$ as 0; typically we use the same architecture for the detector network as the discriminator when training the GAN, although we make small modifications. After training $D_\theta$, given a candidate point $x$, the membership inference score for $x$ is the predicted probability under $D_\theta$ of the point being generated from the GAN, which we'll denote $D_\theta(x)$. The high-level pipeline described above for training $D_\theta$ is shown in Figure 3, with the algorithmic details of how to set the hyper-parameters deferred to the relevant experimental sections.

The variant of the Detector attack we call the Augmented Detector (ADIS) builds on the Detector by augmenting the feature space when training $D_\theta$ with additional statistics beyond just the sample itself. The pipeline is similar to that outlined in Figure 3 with the following additional steps: (i) Sample without replacement, $N(\approx 200)$ data point from the reference and synthetic sample data in Figure 3 respectively. The sub-sampled data are for the computation of the reconstruction losses used to augment the feature space (see Section 4.2) and are not included again in the downstream training set (ii) For each point $x$ in the training set of $D_\theta$, use the dataset sampled in (i) to compute the distance-based test statistics and the DOMIAS likelihood ratio statistic of van Breugel et al. (2023b). The full details of how we train ADIS are deferred to Appendix D.

Now let $f : \mathcal{X} \to [0, 1]$, and $\text{Lip}(f)$ denote the Lipschitz constant of $f$. We can relate the error of $f$ when used for membership inference on points sampled from $\mathcal{R}$ to the error of $f$ when used to classify whether points from $\mathcal{M}$ were sampled from $\mathcal{G}$ or $\mathcal{P}$. If points sampled from $\mathcal{P}$ are labeled 0 and points from $\mathcal{G}$ are labeled 1, then the FPR of $f$ used to classify points from $\mathcal{M}$ is $\text{FPR}_{\mathcal{M}} = \frac{1}{2}\mathbb{E}_{\mathcal{P}}[f(x)]$. Now, the FPR of $f$ as an MIA is $\text{FPR}_{\mathcal{R}} = \mathbb{E}_{\mathcal{R}}[f(x)|x \in \mathcal{P}] = \text{FPR}_{\mathcal{M}}$, so the FPRs are the same. The TPR is more interesting; it is easy to show (Lemma A.1) that

$$\text{TPR}_{\mathcal{R}}(f) \leq \text{TPR}_{\mathcal{M}}(f) + \frac{\text{Lip}(f)}{2}W^1(\mathcal{G}, \mathcal{T}), \tag{1}$$

where $W^1(\mathcal{G}, \mathcal{T})$ is the 1-Wasserstein distance between $\mathcal{G}$ and $\mathcal{T}$. Note that when $\mathcal{G} = \mathcal{T}$ then $\mathcal{R} = \mathcal{M}$, and the above inequality becomes equality, with $W^1(\mathcal{G}, \mathcal{T}) = 0$. Equation 1 motivates the Detector attack, because although we can't directly sample from $\mathcal{R}$, and therefore can't train $f$ to maximize $\text{TPR}_{\mathcal{R}}(f)$ at a fixed FPR rate, in the black-box setting we can still sample from $\mathcal{M}$, and so we can train $f$ that maximizes $\text{TPR}_{\mathcal{M}}(f)$ subject to $\text{FPR}_{\mathcal{M}}(f) = \text{FPR}_{\mathcal{R}}(f) = \alpha$ for any $\alpha \in [0, 1]$. While Equation 1 is nice in that we can upper bound the rate we'd expect $\text{TPR}_{\mathcal{R}}(f) \to \text{TPR}_{\mathcal{M}}(f)$ as $\mathcal{G} \to \mathcal{T}$, namely the rate $W^1(\mathcal{G}, \mathcal{T}) \to 0$, it doesn't tell us much about why in practice a classifier $f$ that minimizes error on $\mathcal{M}$ might achieve low error on $\mathcal{R}$: (i) $f$ is a neural network and so $\text{Lip}(f)$ could be very large in practice, making Equation 1 vacuous and (ii) $W^1(\mathcal{G}, \mathcal{T})$ can be quite large.

In Theorem 4.1 we prove the neat result that if $\mathcal{G}$ is a mixture of the training set distribution $\mathcal{T}$ and the dataset $\mathcal{P}$, and $f^*$ is the Bayes optimal classifier with respect to $\mathcal{M}$, then the MIA that thresholds based on $f^*$ achieves maximal TPR at any fixed FPR on the distribution $\mathcal{R}$. We defer the proof to the Appendix, which uses the form of $\mathcal{G}$ to decompose the error on $\mathcal{G}$ into errors on $\mathcal{T}, \mathcal{P}$, and follows from the Neyman Pearson lemma Neyman & Pearson (1933) and the assumption that $f^*$ is Bayes optimal.

**Theorem 4.1.** *Suppose that $\mathcal{G} = \beta\mathcal{P} + (1 - \beta)\mathcal{T}$ for some $\beta \in [0, 1]$. Let $f^* : \mathcal{X} \to [0, 1]$ be the posterior probability a sample from $\mathcal{M}$ is drawn from $\mathcal{G} : f^*(x) = \frac{\mathcal{G}(x)}{\mathcal{G}(x)+\mathcal{P}(x)}$. Then for any fixed $\alpha \in [0, 1]$, $\exists \tau_\alpha$ such that attack*

$$F^*(x) = \boldsymbol{1}\{f^*(x) > \tau_\alpha\}$$

*satisfies $\text{FPR}_{\mathcal{R}}(F^*) = \alpha$, and for any MIA $f$ such that $\text{FPR}_{\mathcal{R}}(f) = \alpha$, $\text{TPR}_{\mathcal{R}}(f) \leq \text{TPR}_{\mathcal{R}}(F^*)$.*

In practice of course we don't have access to the densities $\mathcal{G}(x), \mathcal{P}$. Rather than approximate the likelihood ratio directly as in van Breugel et al. (2023b), we observe that as long as (i) the hypothesis class $\mathcal{F}$ is sufficiently rich that $f^*(x)$ is closely approximated by a function in $\mathcal{F}$, standard results on the consistency of the MLE show that $f^* = \arg\min_{f \in \mathcal{F}} -\mathbb{E}_{x \sim \mathcal{G}}[\log f(x)] - \mathbb{E}_{x \sim \mathcal{P}}[\log 1 - f(x)]$, and so we can compute $f^*$ implicitly by minimizing the binary cross-entropy loss classifying samples from $\mathcal{M}$! Note that we aren't restricted to use the negative log-likelihood, any Bayes consistent loss function Lin (2004) will suffice as a surrogate loss for the classification step; we use negative log-likelihood in our experiments. The assumption $\mathcal{G}$ is a simple mixture distribution is a stronger assumption, and Theorem 4.1 is better viewed as showing why our Detector outperforms the random baseline by a significant margin, but is likely far from the information theoretically optimal MIA. We note also that the widely observed phenomenon of partial mode-collapse Thanh-Tung et al. (2018), the tendency of GANs in particular to regurgitate a subset of their training data rather than learning the entire distribution, is another motivation for expressing $\mathcal{G}$ as a mixture of $\mathcal{T}$ and $\mathcal{P}$.

## 4.2 DISTANCE-BASED ATTACKS.

The distance-based black-box attacks described in this section are based on the work of Chen et al. (2020). The intuition behind the attack is that points $\tau$ that are closer to the synthetic sample than to the reference samples are more likely to be classified as being part of the training set. We define 3 reconstruction losses that we use to define 2 types of distance-based attacks. Given a distance metric $\delta$, a test point $\tau \sim \mathcal{R}$, and samples $X_G \sim \mathcal{G}, |X_G| = n$ from the target GAN, we define the generator reconstruction loss $R(\tau|X_G) = \arg\min_{s \in X_G} \delta(s_i, \tau)$. Similarly, given a distance metric $\delta$, a test point $\tau$ and reference samples $X_R \sim \mathcal{P}$, we define the reference reconstruction loss $R_{ref}(\tau|X_R) = \arg\min_{x \in X_R} \delta(x, \tau)$. Then we can define the relative reconstruction loss, $R_L(\tau|X_G, X_R) = R(\tau|G) - R_{ref}(\tau|X_R)$. In the *two-way distanced-based* MI attack, we compute the relative reconstruction loss,

$R_L(\tau|G, X_R)$, and predict $\tau \in X_T$ if $x > \lambda$ for some threshold $\lambda$. In the *one-way distanced-based* MI attack, we substitute the relative reconstruction loss with the *reconstruction loss relative to the GAN*, $R_L(\tau|G)$. We illustrate the two-way and one-way distance based attacks in Figures 4, 5 in the Appendix respectively.

Given a set $\Delta$ of distance metrics, for example hamming and Euclidean distance, we can naturally define a weighted distance-based attack that computes a weighted sum of the reconstruction losses for each distance metric, which we define in Equation 2 in the Appendix. In the genomic data setting, we also implement the distance-based attack of Homer et al. (2008b) that is tailored to attacks on genomic data. An improved "robust" version of this attack was proposed in Dwork et al. (2015). The detailed outline of the robust version of this attack is presented in Algorithm 1 in the Appendix.

## 5 ATTACKS ON TABULAR DATA GANS

In this section we discuss MIA attack success against a variety of GAN architectures trained on tabular data. We evaluate 7 attacks across datasets of dimension $d = 805, 5000, 10000$. We first discuss the genomic datasets we used to train our target GANs and subsequently evaluate our attack methods.

### 5.1 EXPERIMENTAL DETAILS

We evaluate our attacks on genomics data from two genomics databases 1000 Genomes (1KG) Consortium et al. (2015) and dbGaP Mailman et al. (2007)). We pre-processed the raw data from dbGaP to Variant Call Format (VCF) Danecek et al. (2011) with PLINK Purcell et al. (2007), we used the pre-processed 1KG data provided by Yelmen et al. (2021) without any further processing. For both datasets we sub-sampled the features to create three datasets of dimensions $\{805, 5000, 10000\}$ by selecting regular interval subsets of the columns. For each sub-sampled dataset, we trained two types of target GANs: the vanilla GAN Goodfellow et al. (2014) using the implementation from Yelmen et al. (2021), and the Wasserstein GAN with gradient policies (WGAN-GP) Gulrajani et al. (2017). For the 1KG dataset we train the target GAN on 3000 samples, and use a further 2008 samples to train the Detector. We evaluate our attacks using another held out sample of 500 test points, and 500 randomly sampled points used to train the target GAN. On dbGaP we train the target GAN, Detector, and evaluate on 6500, 5508, 1000 data points respectively. Table 1 in the Appendix summarizes this setup. Figures $7a - 7f$ depict the top 6 principal components of both the training data and synthetic samples for each configuration in Table 1, showing visually that our GAN samples seem to converge in distribution to the training samples. We further verify that our GANs converged and did not overfit to the training data in Appendix F.2.1. In total, 7 attack models (*one-way distance*, *two-way distance*, *robust homer*, *weighted distance*, *Detector*, *ADIS*, *DOMIAS*) were evaluated on different genomic data configurations. Each of our distance attacks we introduce in Section 4.2 requires a further choice of distance metric, $\delta$, that might be domain-specific and influenced by the nature of the data. We tested our attacks using Hamming distance and Euclidean distance, finding that Hamming distance performs better across all settings, and so we use that as our default distance metric. For the Robust Homer attack, in order to compute the synthetic data mean and reference data means we use all of the reference samples except one held-out sample, and an equal number of samples from the generator. We defer further details on detector architecture and training hyperparameters to Appendix F.3. Table 2 Appendix C lists the Detector's test accuracy on $\mathcal{M}$, verifying we are successful in training the Detector. The details for ADIS training are discussed in detail in Appendix D.

### 5.2 MIA SUCCESS ON GENOMIC GANS

We now discuss the relative attack success of our methods, reporting full ROC curves plotted on the log-log scale (Figures 1, 1f), as well as Table 3 which reports the TPR at fixed low FPR rates for each attack. For both datasets, we trained average the ROC Curves and TPR/FPR results over 11 training runs where each time we trained a new target GAN on a new sub-sample of the data.

Inspecting the results on 1KG in Figure 1, we see that when the target GAN is the Vanilla GAN, at 805 SNPs the two-way distance attack achieves both the highest TPRs at low FPRs, and the highest AUC overall at .66, with all the attacks except the one-way distance attack outperforming the random

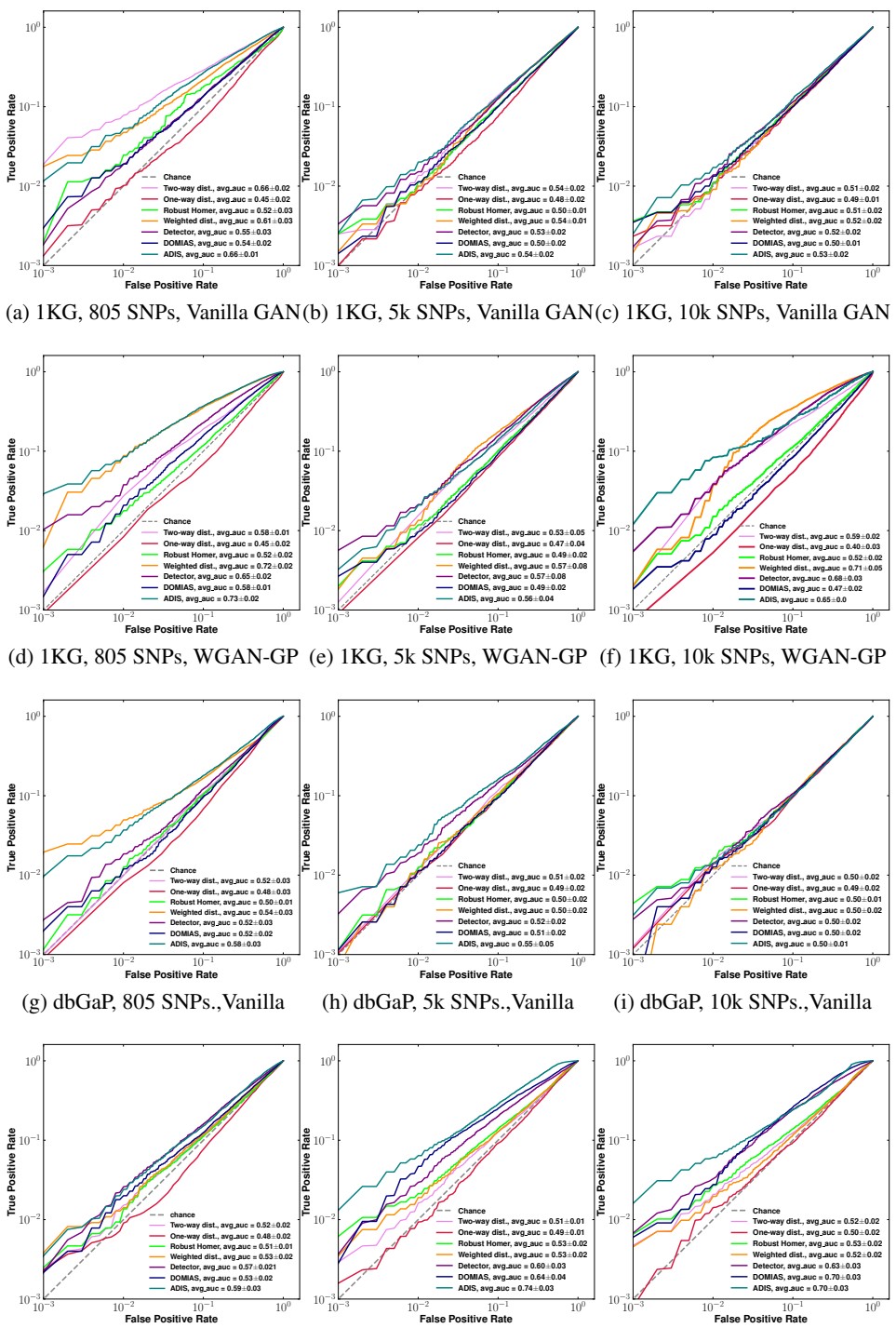

Figure 1: log-log ROC Curves for Vanilla GAN and WGAN-GP trained on 1KG and dbGaP. Results are averaged over 11 runs, with the mean AUC $\pm$ the standard deviation reported in the legend.

baseline (curves above the diagonal) by a significant margin. As the number of SNPs increases to $5K$ and then $10K$, we see that the two-way distance attack performs worse than ADIS and the Detector, even performing worse than random guessing at $10K$. As the number of SNPs increases, the relative performance of the Robust Homer Attack increases, which we also observe in the WGAN-GP results, and on the dbGaP dataset. When the target GAN is a WGAN-GP, detector-based attacks outperform distance-based attacks or DOMIAS at every SNP dimension, which we also observe for the dbGaP dataset. While the trends in terms of relative performance of different attack methods are consistent across datasets, the actual levels of privacy leakage differ. Inspecting the dbGaP results in Figure 1f we see lower AUCs across the board, although we still observe high TPRs at low FPRs. Interestingly, the only method that consistently fails to beat the random baseline (AUC $< .5$) in all settings is the one-way distance-based attack. Paired with the moderate success of two-way distance attacks, this shows that for training points while the relative difference between their distance to generated samples and their distance to test samples is larger than the difference in distances for test points, but the actual distance to the generated samples alone is not uniformly larger.

More generally, while the AUC results for the attack methods indicate moderately low privacy leakage relative to white-box attacks against the discriminator Hayes et al. (2018); Chen et al. (2020); Hilprecht et al. (2019) or those reported against diffusion models Carlini et al. (2023), for the more meaningful metric of FPR at low fixed TPRs Carlini et al. (2021), Table 3 shows that for fixed FPRs ADIS in particular achieves TPRs that are as much as 10x the random baseline (FPR = TPR). ADIS works especially well against WGAN-GP trained on dbGaP, with improved attack success for larger dimension.

## 6    ATTACKS ON IMAGE GANS

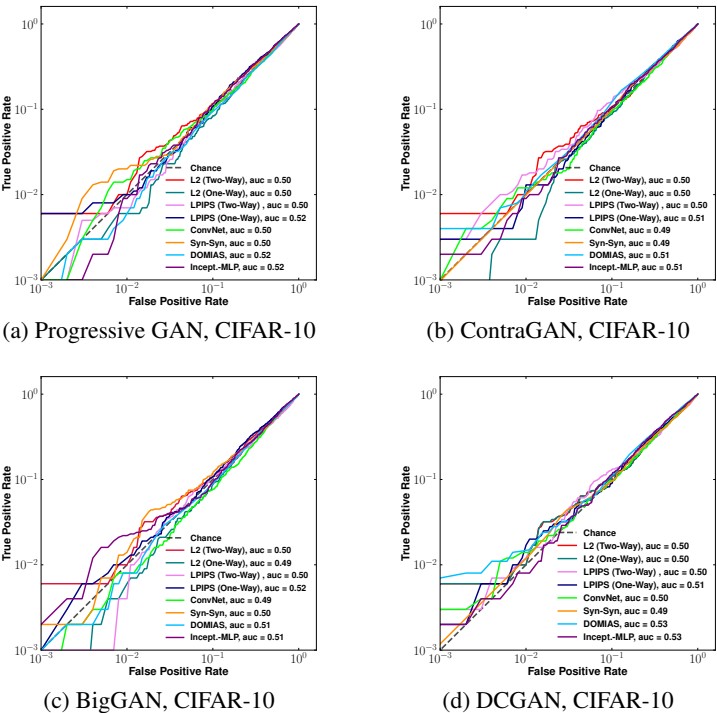

(a) Progressive GAN, CIFAR-10    (b) ContraGAN, CIFAR-10

(c) BigGAN, CIFAR-10    (d) DCGAN, CIFAR-10

Figure 2: log-log ROC curves for detector-based MIAs against different GAN architectures trained on CIFAR-10. While the AUCs are lower than the genomic settings, the different variants of the detector-based attacks work very well at low FPRs.

In this Section, we evaluate our attacks against 4 GAN architectures trained on images from the CIFAR-10 dataset. We observe that relative to high dimensional tabular datasets, image GANs exhibit less privacy leakage on average, with AUCs that are barely above the random baseline. However,

for the most meaningful metric, TPR at low FPRs, our detector-based attacks, and in some cases the distance-based attacks, are again able to achieve TPRs that are $2 - 6\times$ higher than the random baseline.

## 6.1 EXPERIMENTAL DETAILS

We focused on GANs trained on the CIFAR-10 image dataset Krizhevsky (2009). CIFAR-10 consists of 60000 images across 10 object categories, and so when sub-sampling the dataset for training the target GAN, Detectors, and evaluation, we performed stratified sampling to ensure balance across categories. The test sample size was set to 2000 such that we have 200 objects from each category, with 50000 images used to train the target GAN, and 9000 used to train the Detectors. Four state-of-the-art GAN variants were trained: large-scale GANs for high fidelity image synthesis Brock et al. (2018), Conditional GANs with Projective Discriminator (PDGAN) Miyato & Koyama (2018), Deep Convolutional Generative Adversarial Networks (DCGAN) Radford et al. (2016), and Contrastive Learning for Conditional Image Synthesis Kang & Park (2021). A summary of the target GAN configurations is depicted in Table 5 in the Appendix. We trained the GANs using the implementations in the Pytorch StudioGAN library Kang et al. (2023a). We observed that training the Detector on image data proved a more delicate process than distinguishing genomic data – as a result we implemented 3 Detector architectures. The first variant trains a CNN-based Detector Krizhevsky et al. (2017); Albawi et al. (2017). In the second variant we extracted features from the training images using a pre-trained model (Inception model architecture) Szegedy et al. (2014) and fed this to an MLP pipeline. In the final variant, we trained a variational auto-encoder (VAE) Kingma & Welling (2022) on the 9000 reference data images, and then trained a CNN classifier Krizhevsky et al. (2017) on synthetic samples from the VAE (label 0) and synthetic samples from the target GAN (label 1). This idea of training a second generative model on the reference data was proposed in Hayes et al. (2018). Further details on Detector architectures, training parameters, and convergence plots are deferred to Section F.1 in the Appendix.

## 6.2 MIA ATTACK SUCCESS

Recall that for the image GANs we have 3 variants of the detector attack, DOMIAS, and two distance measures ($\ell_2$, LPIPS Zhang et al. (2018)) were utilized for the distance-based attack. Each distance measure contributes 2 attack types: one-way and two-way, and so for the full results in the Appendix we have a total of 8 attacks. We summarize the results in Figure 2 and Table 3 We see that at low FPRs $< 10^{-2}$, the Detector that achieves the highest TPR varies based on the target GAN architecture. For all of the target GANs Incept-MLP has the highest AUC of the detector-based methods, but performs poorly at low FPRs, albeit on BigGAN, where its AUC is about the same as the other methods. DOMIAS performs better on a relative basis on the image GANs, with the best performance in terms of AUC and low FPRs on DCGAN and second best performance at low FPRs on ContraGAN. However, when the target GAN was Progressive GAN or BigGAn, DOMIAS performed worse than random guessing at low FPRs, and so its performance overall seems inconsistent. This is an interesting direction for future investigation.

## 7 CONCLUSION

Our work provides the most thorough existing empirical analysis of attacks against GANs in the black-box setting. We prove theoretical results that build intuition for why a network that can detect samples generated from the GAN can also detect samples from the training set (Theorem 4.1), a result that has applications across all privacy attacks on ML models, not just GANs, and conduct an extremely thorough empirical evaluation of detector-based attacks, comparing their performance to existing black-box attacks across tabular and image domains. This work also raises several interesting directions for future research: although we expose significant privacy leakage from GANs, particularly at low FPRs, relative to other generative models explored in prior work, for example VAEs Hayes et al. (2019) or diffusion models Carlini et al. (2023), black-box access to the generator seems much more private. Is this actually the case, or is it simply a matter of developing better attack methods? If so, can we prove theoretically why GANs appear to be more privacy-preserving than other generative models? Future work will also explore scaling DP GANs to higher dimensions in order to provide theoretical defense guarantees against MIA attacks.

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

## A    LEMMAS AND PROOFS

**Lemma A.1.**

$$TPR_{\mathcal{R}} \geq TPR_{\mathcal{M}}(f) - \frac{Lip(f)}{2}W^1(\mathcal{G}, \mathcal{T}),$$

*Proof.* $\mathrm{TPR}_{\mathcal{R}}(f) = \frac{1}{2}\mathbb{E}_{\mathcal{T}}[f(x)] = \frac{1}{2}(\mathbb{E}_{\mathcal{G}}[f(x)] + \mathbb{E}_{\mathcal{T}}[f(x)] - \mathbb{E}_{\mathcal{G}}[f(x)]) = \mathrm{TPR}_{\mathcal{M}}(f) + \frac{1}{2}(\mathbb{E}_{\mathcal{T}}[f(x)] - \mathbb{E}_{\mathcal{G}}[f(x)]) \geq \mathrm{TPR}_{\mathcal{M}}(f) - \frac{1}{2}(|\mathbb{E}_{\mathcal{T}}[f(x)] - \mathbb{E}_{\mathcal{G}}[f(x)]|) \geq \mathrm{TPR}_{\mathcal{M}}(f) - \frac{1}{2}(\mathrm{Lip}(f)\sup_{g:\mathrm{Lip}(g)=1}|\mathbb{E}_{\mathcal{T}}[g(x)] - \mathbb{E}_{\mathcal{G}}[g(x)])|) = \mathrm{TPR}_{\mathcal{M}}(f) - \frac{\mathrm{Lip}(f)}{2}W^1(\mathcal{G}, \mathcal{T}).$ $\square$

**Proof of Theorem 4.1**

*Proof.* Let $f : \mathcal{X} \to \{0, 1\}$ be an arbitrary MIA achieving FPR $\alpha$, that is $\mathbb{E}_{x \sim \mathcal{D}}[f(x)] = \alpha$. Then the TPR of $f$ at distinguishing samples from $G$ is $\mathbb{E}_{x \sim G}[f(x)] = \beta\mathbb{E}_{x \sim \mathcal{T}}[f(x)] + (1 - \beta)\mathbb{E}_{x \sim \mathcal{D}}[f(x)]$, since $G = \beta\mathcal{T} + (1 - \beta)\mathcal{D}$. Substituting in the FPR of $f$, we have $\mathbb{E}_{x \sim G}[f(x)] = \beta\mathbb{E}_{x \sim \mathcal{T}}[f(x)] + (1 - \beta)\alpha$. But $\mathbb{E}_{x \sim \mathcal{T}}$ is just the TPR at detecting samples from $\mathcal{T}$. We can take $\inf_{f:\mathrm{FPR}(f)=\alpha}$ of both sides of the prior equation, which shows that the MIA attack achieving optimal TPR at a fixed FPR $\alpha$ is exactly the optimal hypothesis test for distinguishing samples from $G$ from samples from $D$ with a fixed FPR $\alpha$. Next we note that the optimal hypothesis test in this latter scenario is characterized by the Neyman-Pearson Lemma Neyman & Pearson (1933): There exists $\tau_\alpha$ such that $f^*(x) = \mathbf{1}\{\frac{G(x)}{D(x)} > \tau_\alpha\}$ achieves the maximum achievable TPR at fixed FPR $\alpha$, and so this is the optimal MIA for detecting training samples from $\mathcal{T}$ at a fixed FPR $\alpha$. It remains to be shown that we recover $f^*$ of this form from minimizing the classification error on $\frac{1}{2}D + \frac{1}{2}G$. Under our assumption that we compute a classifier that exactly minimizes the loss on the distribution, it is a standard result that the optimal classifier is the Bayes optimal classifier $B(x) = \frac{G(x)}{G(x)+D(x)}$. Then the result follows from noting that $B(x) > \tau$ is equivalent to $\frac{G(x)}{D(x)} > \frac{1}{\frac{1}{\tau}-1}$. $\square$

We first present pseduocode for the Robust Homer attack, Detector attack and ADIS.

## B    ROBUST HOMER ATTACK

The intuition behind the attack lies in Line 7 of Algorithm 1. Recall $X_G$ denotes synthetic samples from the GAN, $X_R$ denotes reference data sampled from $\mathcal{P}$, and $\tau \in \mathcal{E}$ denotes a test point. Let $\mu_g = \frac{1}{n}\sum_{i=0}^{n} s_i$ where $x_{g0}, x_{g1}, \ldots x_{gn} \in X_G$ and $\mu_g \in [0,1]^d$. Similarly, let $\mu_r = \frac{1}{m}\sum_{i=1}^{m} x_i$ where $x_1, x_2 \ldots x_m \in X_R$ and $\mu_r \in [0,1]^d$. Then sample $x \sim \mathcal{D}$, and compute:

$$\langle \tau - x, \mu_g - \mu_r \rangle = \langle \tau, \mu_g - \mu_r \rangle - \langle x, \mu_g - \mu_r \rangle = \underbrace{\left[\langle \tau, \mu_g \rangle - \langle x, \mu_g \rangle\right]}_{(1)} + \underbrace{\left[\langle x, \mu_r, \rangle - \langle \tau, \mu_r, \rangle\right]}_{(2)}$$

Part (1) above checks if $\tau$ is more correlated with $\mu_g$ than a random sample $x$ from $\mathcal{D}$, while (2) checks if $x$ is more correlated with $\mu_r$ than $\tau$. Observe that this is similar to the intuition behind the distance-based attack, but here we compute an average *similarity measure* to $\mu_g$ and $\mu_r$, rather than a distance to the closest point.

## C    DETECTOR ATTACK

Figure 3 illustrates the Detector pipeline. The target GAN configurations for the genomic data setting is shown in Table 1:

---

**Algorithm 1** Robust Homer Attack

---

**Require:** $(X_R \in \{0,1\}^d, G, \mathcal{E}, x)$

1: $\mu_r = \frac{1}{m}\sum_{i=1}^{m} x_i$    where    $x_1, x_2 \ldots x_m \in X_R$

2: $\alpha \leftarrow \frac{1}{\sqrt{m}} + \epsilon, \quad \eta \leftarrow 2\alpha$                                $\triangleright \epsilon \geq 0$

3: $X_G \sim G$                                      $\triangleright X_G \in \{0,1\}^d$

4: $\mu_g = \frac{1}{n}\sum_{i=0}^{n} s_i$    where    $x_{g0}, x_{g1}, \ldots x_{gn} \in X_G$    and    $\mu_g \in [0,1]^d$

5: Let $\lfloor \mu_g - \mu_r \rceil_\eta \in [-\eta, +\eta]^d$          $\triangleright$ entry-wise truncation of $\mu_g - \mu_r$ to $[-\eta, \eta]$

6: **for** $\tau \in \mathcal{E}$ **do**

7:      $\rho \leftarrow \langle \tau - x, \lfloor \mu_g - \mu_r \rceil_\eta \rangle$

8:      **if** $\rho > \kappa$ **then**                           $\triangleright \kappa$ is an hyperparameter

9:          $\tau$ is in training set for $G$

10:      **else**

11:          $\tau$ is not in training set for $G$

12:      **end if**

13: **end for**

---

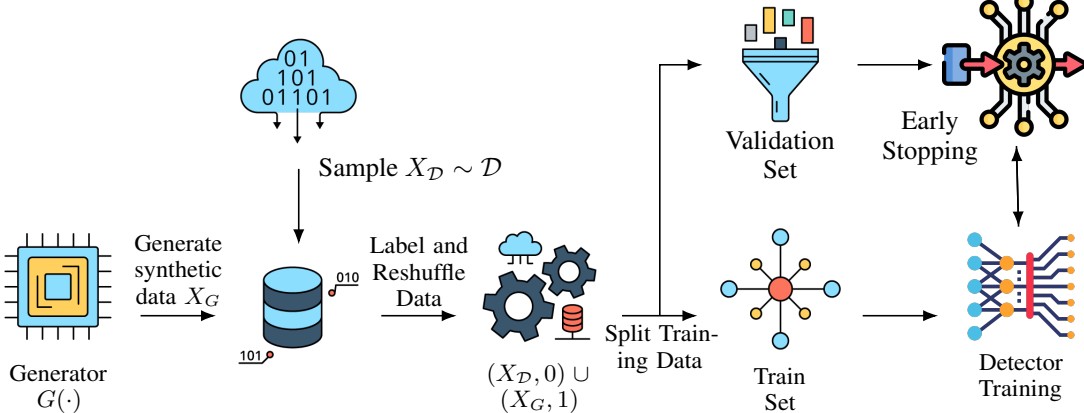

Figure 3: The pipeline for training the Detector $D_\theta$.

Table 1: Target GAN configurations for genomic Data

| Data | SNPs Dim. | Train Data Size | Ref. Data Size | Test Size | GAN Variant |
|---|---|---|---|---|---|
| 1000 Genome | 805 | 3000 | 2008 | 1000 | vanilla & WGAN-GP |
| | 5000 | 3000 | 2008 | 1000 | vanilla & WGAN-GP |
| | 1000 | 3000 | 2008 | 1000 | vanilla & WGAN-GP |
| dbGaP | 805 | 6500 | 5508 | 1000 | vanilla & WGAN-GP |
| | 5000 | 6500 | 5508 | 1000 | vanilla & WGAN-GP |
| | 10000 | 6500 | 5508 | 1000 | vanilla & WGAN-GP |

---

**Algorithm 2** Detector

---

**Require:** $\{f_\theta, X_R, G, \mathcal{E}\}$ ▷ $f_\theta$=multilayer perceptron , $X_R$=reference data, $G(\cdot)$= black box GAN,
   $\mathcal{E}$ = test samples.
1: $epoch := 150$
2: $batch\_size := 100$
3: $lr := $ 1e-3          ▷ learning rate
4: $Optimizer := Adam$          ▷ Optimizer for training
5: $f_s =$ EARLYSTOP()          ▷ early stopping routine
6: $f_{lr} =$ DECAYRATE$(lr)$          ▷ decaying learning rate
7: **for** $j = 0$ to $epoch - 1$ **do**
8:      **if** $j \bmod 3 = 0$ **then**
9:          $X_G \sim G$          ▷ sample synthetic data from GAN
10:          $y_{X_R} \leftarrow \mathbf{0}$          ▷ label (reference sample)
11:          $y_S \leftarrow \mathbf{1}$          ▷ label (synthetic sample)
12:          $\mathbf{X} = \{(X_G, y_G) \cup (X_R, y_{X_R})\}$          ▷ Combine data
13:          $\mathbf{X}_T, \mathbf{X}_V =$ SPLIT$(\mathbf{X})$          ▷ Split data into training & validation sets
14:      **end if**
15:      $f_\theta \leftarrow f_\theta(\mathbf{X}_T, \mathbf{X}_V, batch\_size, f_{lr}, f_s, Optimizer)$          ▷ model training and update
16: **end for**
17: $output \leftarrow f_\theta(\mathcal{T})$          ▷ predict test samples

---

**Detector Prediction Score.** We might ask how well does the Detector performs in classifying synthetic samples as a first step in checking how well-trained is. Table 2 shows that the Detector predictions are very high for synthetic samples as expected. However, the goal is for the Detector to be able to distinguish training and non-training samples in the test set, given that both samples are from the same underlying distribution.

Table 2: Detector Test Accuracy

| Data | GAN Variant | SNPs Dim. | Test Size | Train Epochs | Mean Accuracy |
|---|---|---|---|---|---|
| 1000 Genome | vanilla | 805 | 1000 | 159 | 0.974 |
| | vanilla | 5000 | 3000 | 240 | 0.980 |
| | vanilla | 1000 | 3000 | 360 | 0.999 |
| dbGaP | WGAN-GP | 805 | 1000 | 300 | 0.995 |
| | WGAN-GP | 5000 | 1000 | 600 | 0.998 |
| | WGAN-GP | 10000 | 6500 | 1500 | 0.930 |

## D   Augmented Detector(ADIS)

**ADIS Training.** The architecture for ADIS is the same as the Detector, but there are additional preprocessing steps and fewer epochs ($\approx$ 10-30 epochs). A fixed sample size of 300 from the

reference and synthetic sample was set aside for the computation of reconstruction losses 4.2. This was followed by fitting principal component analysis (PCA) Bro & Smilde (2014); Ringnér (2008) on the training data (consisting of reference and synthetic data) and selecting the first 100-300 principal components - the actual component selected depends on the dimension of the original feature space, but typically 100 components were selected for 805 SNPs and 300 for 5000 SNPs. Subsequently, we computed *one-way and two-way* hamming reconstruction losses 4.2 on the non-transformed training data. Furthermore, as recommended in Hilprecht et al. (2019), distance attacks also work well on reduced feature space. Thus, we computed the *two-way* L2 reconstruction loss on the PCA-transformed training data. Note that the 300 samples that was set aside had to be PCA-transformed before being used for the computation of *two-way* L2 reconstruction loss. For the computation of the test statistics of DOMIAS van Breugel et al. (2023b), we first reduced the dimensionality of the synthetic samples and reference data separately with the fitted PCA. Then we fitted a Gaussian mixture model Reynolds et al. (2009), $P_G$, on the PCA-transformed synthetic samples and another Gaussian mixture model, $P_{X_R}$, on the PCA-transformed reference data. Finally, we computed the test statistic $\frac{P_G(x)}{P_{X_R}(x)}$ for each PCA-transformed training point $x$.

**Large Reference Data for ADIS**   It should be noted that the Detector and ADIS attacks would benefit from having a reference data sample of large size. In cases where the reference data size is small, an effective strategy would be to train a secondary GAN on the reference data. Subsequently, one can proceed to subsample synthetic reference data (from the trained secondary GAN) that are closer or within an $\epsilon$-ball of the reference data as measured by a distance metric - i.e., using distance-based subsampling.

**Results Table for GAN-Omics MIA**   Table 3 depicts the TPR at FPR of $0.01$ , $0.1, 0.005$ and $0.001$ respectively (see Figures 1 and 1d) for genomic data setting.

**Training DOMIAS**   Training DOMIAS involves 2 steps - dimensionality reduction and density estimation. For genomic data setting, we use PCA for dimension reduction following the same steps as described for ADIS. For density estimation, we fit density estimator using non-volume preserving transformation(NVP)Dinh et al. (2016). Observe, that we fit the the estimator to the synthetic and reference sample separately and then proceed to compute the likelihood ratio for any given test point. For image data, we project the image onto a low dimensional space using the encoder of a variational auto-encoderKingma & Welling (2022) trained on the reference and synthetic data. Subsequently, we fit the density estimator using NVP to the transformed data.

# E   DISTANCE-BASED ATTACKS

## E.1   WEIGHTED DISTANCE ATTACK

Given a set $\Delta$ of distance metrics, for example hamming and Euclidean distance, we can naturally define a weighted distance-based attack that computes a weighted sum of the reconstruction losses for each distance metric:

$$\mathcal{W} = \sum_{\delta \in \Delta} \alpha_\delta (R^\delta(\tau|G) - R^\delta_{ref}(\tau|X_R)) = \sum_{\delta \in \Delta} \alpha_\delta (R^\delta_L(\tau|G, X_R)) \qquad (2)$$

The notations $R^\delta(\tau|G)$ and $R^\delta_{ref}(\tau|X_R)$ emphasize dependence on the distance metric. The weighting parameter $\alpha_\delta$ is a measure of how important the associated distance metric is for membership inference, and could be selected based on domain knowledge.

Figures 5 and 5 illustrates the intuition behind the one-way and two-way distance-based attack.

# F   RESULTS FOR GENOMIC GANS

## F.1   GENOMIC GAN CONVERGENCE PLOTS

For image data, visual inspection is a quick way to examine if the synthetic samples converge to the underlying real training samples. Clearly, under the genomic tabular setting, such quick visual convergence examination does not work. Recent work of Platzer (2013) proposed using both the

Table 3: Table of Attack Results (Genomic Data)

| GAN Variant | Dataset | Attack Method | TPR @0.01 FPR | TPR @ 0.1 FPR | TPR @0.005 FPR | TPR @0.001 FPR |
|---|---|---|---|---|---|---|
| vanilla 805 | 1000 genome | one-way distance | 1.02% | 6.95% | 0.51% | 0.13% |
| vanilla 805 | 1000 genome | two-way distance | 7.79% | 28.86% | 5.14% | 1.89% |
| vanilla 805 | 1000 Genome | weighted distance | 4.75% | 21.76 % | 2.85% | 1.76% |
| vanilla 805 | 1000 Genome | Robust Homer | 2.46% | 17.89% | 1.26% | 0.20% |
| vanilla 805 | 1000 Genome | Detector | 1.89% | 13.72% | 0.97% | 0.19% |
| vanilla 805 | 1000 Genome | ADIS | 5.31% | 26.89% | 3.14% | 1.16% |
| vanilla 5k | 1000 genome | one-way distance | 0.88% | 7.6% | 0.37% | 0.10% |
| vanilla 5k | 1000 genome | two-way distance | 1.36% | 13.89% | 0.68% | 0.25% |
| vanilla 5k | 1000 Genome | weighted distance | 1.00% | 12.6% | 0 .56% | 0.15% |
| vanilla 5k | 1000 Genome | Robust Homer | 0.88% | 7.62% | 0.58% | 0.25% |
| vanilla 5k | 1000 Genome | Detector | 1.52% | 12.70% | 0.82% | 0.33% |
| vanilla 5k | 1000 Genome | ADIS | 2.00% | 13.40% | 0.99% | 0.26% |
| vanilla 10k | 1000 genome | one-way distance | 1.10% | 9.97% | 0.62% | 0.23% |
| vanilla 10k | 1000 genome | two-way distance | 1.07% | 10.52% | 0.42% | 0.17% |
| vanilla 10k | 1000 Genome | weighted distance | 0.90% | 10.4% | 0.49% | 0.15% |
| vanilla 10k | 1000 Genome | Robust Homer | 1.11% | 10.56% | 0.58% | 0.36% |
| vanilla 10k | 1000 Genome | Detector | 1.38% | 11.56% | 0.68% | 0.17% |
| vanilla 10k | 1000 Genome | ADIS | 1.72% | 12.56% | 1.03% | 0.25% |
| wgan-gp 805 | dbGaP | one-way distance | 0.91% | 7.69% | 0.54% | 0.23% |
| wgan-gp 805 | dbGaP | two-way distance | 1.52% | 11.91% | 0.63% | 0.23% |
| wgan-gp 805 | dbGaP | weighted distance | 1.41% | 11.75% | 0.91% | 0.38% |
| wgan-gp 805 | dbGaP | Robust Homer | 1.37% | 11.22% | 0.67% | 0.25% |
| wgan-gp 805 | dbGaP | Detector | 2.59% | 16.01% | 1.14% | 0.22% |
| wgan-gp 805 | dbGaP | ADIS | 2.40% | 15.15% | 1.04% | 0.35% |
| wgan-gp 5k | dbGaP | one-way distance | 0.93% | 9.17% | 0.23% | 0.16% |
| wgan-gp 5k | dbGaP | two-way distance | 1.62% | 12.81% | 0.77% | 0.29% |
| wgan-gp 5k | dbGaP | weighted distance | 1.90% | 12.90% | 1.90% | 0.35% |
| wgan-gp 5k | dbGaP | Robust Homer | 2.13% | 14.03% | 1.48% | 0.62% |
| wgan-gp 5k | dbGaP | Detector | 2.95% | 20.10% | 1.64% | 0.37% |
| wgan-gp 5k | dbGaP | ADIS | 6.40% | 28.70% | 3.98% | 1.32% |
| wgan-gp 10k | dbGaP | one-way distance | 1.42% | 9.38% | 0.55% | 0.09% |
| wgan-gp 10k | dbGaP | two-way distance | 2.07% | 12.45% | 1.20% | 0.47% |
| wgan-gp 10k | dbGaP | weighted distance | 1.96% | 11.80% | 0.98% | 0.45% |
| wgan-gp 10k | dbGaP | Robust Homer | 2.71% | 13.95% | 1.35% | 0.65% |
| wgan-gp 10k | dbGaP | Detector | 3.26% | 24.23% | 2.11% | 0.67% |
| wgan-gp 10k | dbGaP | ADIS | 6.07% | 24.49% | 4.24% | 1.6% |

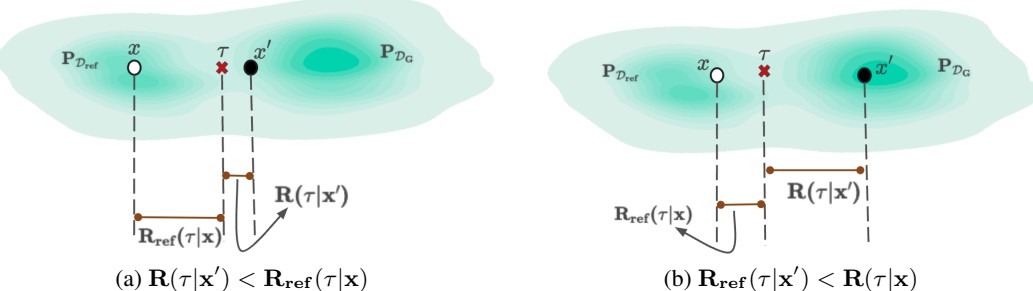

(a) $\mathbf{R}(\tau|\mathbf{x}') < \mathbf{R_{ref}}(\tau|\mathbf{x})$      (b) $\mathbf{R_{ref}}(\tau|\mathbf{x}') < \mathbf{R}(\tau|\mathbf{x})$

Figure 4: The diagram illustrates how the distance-based attack can be used to classify a test point $\tau$. $\mathbf{P}_{\mathcal{D}_{\mathbf{ref}}}$ is exactly the reference distribution, $\mathcal{P}$, while $\mathbf{P}_{\mathcal{D}_{\mathbf{G}}}$ is exactly $\mathcal{G}$, the distribution of synthetic samples from GAN. $X_R \sim \mathcal{P}$ and $x \in X_R, x' \in X_G$ where $X_G \sim \mathcal{G}$. For $(a)$, since $\mathbf{R}(\tau|\mathbf{x}') < \mathbf{R_{ref}}(\tau|\mathbf{x})$, $\tau$ is closer to the synthetic samples and is most likely to be categorized as being from the underlying training set- being closer to synthetic samples implies being closer to the training data since synthetic samples are approximations of the training data. For $(b)$, since $\mathbf{R_{ref}}(\tau|\mathbf{x}) < \mathbf{R}(\tau|\mathbf{x}')$, $\tau$ is closer to $X_R$ and more likely to be classified as not being in the training set for the GAN i.e $\tau \in \mathcal{D}_{ref}$.

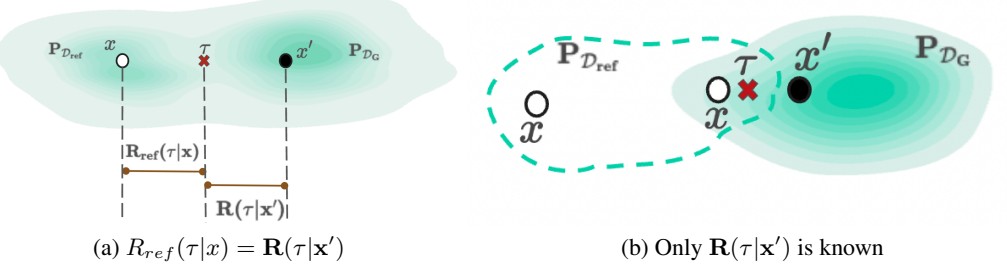

(a) $R_{ref}(\tau|x) = \mathbf{R}(\tau|\mathbf{x}')$      (b) Only $\mathbf{R}(\tau|\mathbf{x}')$ is known

Figure 5: Similar to figure 4, $x \in X_R$, $x' \in X_G$ where $X_R$, $X_G$ are reference and synthetic samples respectively and $\tau$ a test point. Figure $(a)$ above shows the situation where $\mathbf{R_{ref}}(\tau|\mathbf{x}) = \mathbf{R}(\tau|\mathbf{x}')$, and we can conclude that the test point $\tau$ has an equal likelihood to be in $\mathcal{D}$ and $\mathcal{D}_{ref}$, thus the reconstruction loss is inconclusive in this case. Fig $(b)$ depicts the case where we only measure $\mathbf{R}(\tau|\mathbf{x}')$, and disregard $\mathbf{R_{ref}}(\tau|\mathbf{x})$. In this situation, information regarding how close the test point, $\tau$, is to the reference sample is unknown. The reference sample point, $x'$, might be closer to the test point $\tau$ than any synthetic sample or farther; this additional information is lost if we compute only $\mathbf{R}(\tau|\mathbf{x}')$.

principal component analysis (PCA) and $t$-distributed stochastic neighbor embedding (T-SNE) plots for the analysis of genomic data population structure. Consequently, we examine the convergence of the synthetic samples using both PCA and T-SNE. In particular, the synthetic samples converge if the population structure as captured by the PCA and T-SNE is similar to that of the corresponding real genomic samples. Figures 6a - 6f depict the 6 principal components of both the training data and synthetic samples for each configuration in table 1. This provides the first insight that the synthetic samples are indeed representative of the underlying data

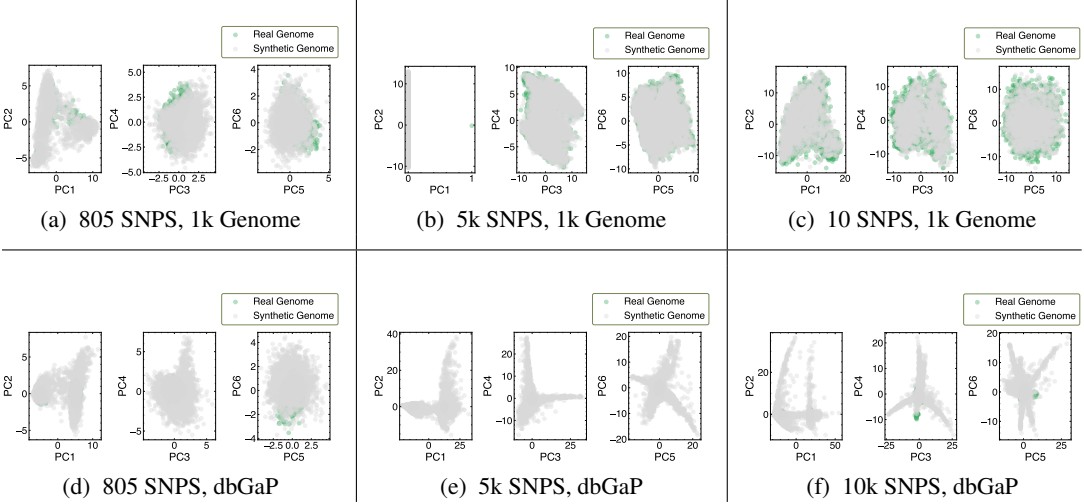

Figure 6: PCA Plots for all data configurations. The plot could be read row-wise or column-wise. The first row depicts PCA plots for synthetic samples from Vanilla GAN trained on 1000 Genome data for 805, 5k, and 10k SNPS respectively. The second row depicts PCA plots for synthetic samples from WGAN trained on dbGaP data for 805, 5k, and 10k SNPs respectively. Each column corresponds to 805,5k and 10k SNPS configurations.

## F.2 OVERFITTING AND MEMORIZATION IN GENOMIC GANS

The section is aimed at performing some preliminary analyses on genomic data to ensure it is not overfitting or memorizing the whole training dataset. It is important to check the effect of these factors are kept to the minimum otherwise they could potentially increase the success margin of the attack schemes.

**Whole sequence memorization.** We iteratively sample batches of 3000-4000 synthetic samples, $S_B$, from the GAN until the sample size is of order $\geq 10^6$. For each sample point in the $i$-th batch, $s_b \in S_{B_i}$, we compute its minimum hamming distance to $\mathcal{D}$, the training data. Observe that a hamming distance of zero would indicate whole sequence memorization - since it implies that an exact copy of a training data sample is being synthesized by the GAN. The results of the whole presented in Figures 9 and 8 memorization showed that the GAN models did not memorize the training data, since no reconstruction loss of zero was encountered- a reconstruction loss of zero reflects that an exact training sample was found.

### F.2.1 OVERFITTING.

Overfitting is helpful for a successful MI attack Yeom et al. (2018). To measure and detect the extent to which GAN overfits the underlying training data we compute the normalized *median recovery error gap* (MRE-gap) Webster et al. (2019). The median recovery errors (MRE) for the training data $X_T$, and subsamples of the reference data, $\mathcal{X}_R \sim \mathcal{D}$, are defined as:

$$MRE_G(X_T) = \text{median}\Big\{ \min_{s_i \in G} \|x_i - s_i\|^2 \Big\}_{x_i \in X_T}$$

$$MRE_G(\mathcal{X}_R) = \text{median}\Big\{ \min_{s_i \in G} \|z_i - s_i\|^2 \Big\}_{z_i \in \mathcal{X}_{ref}}$$

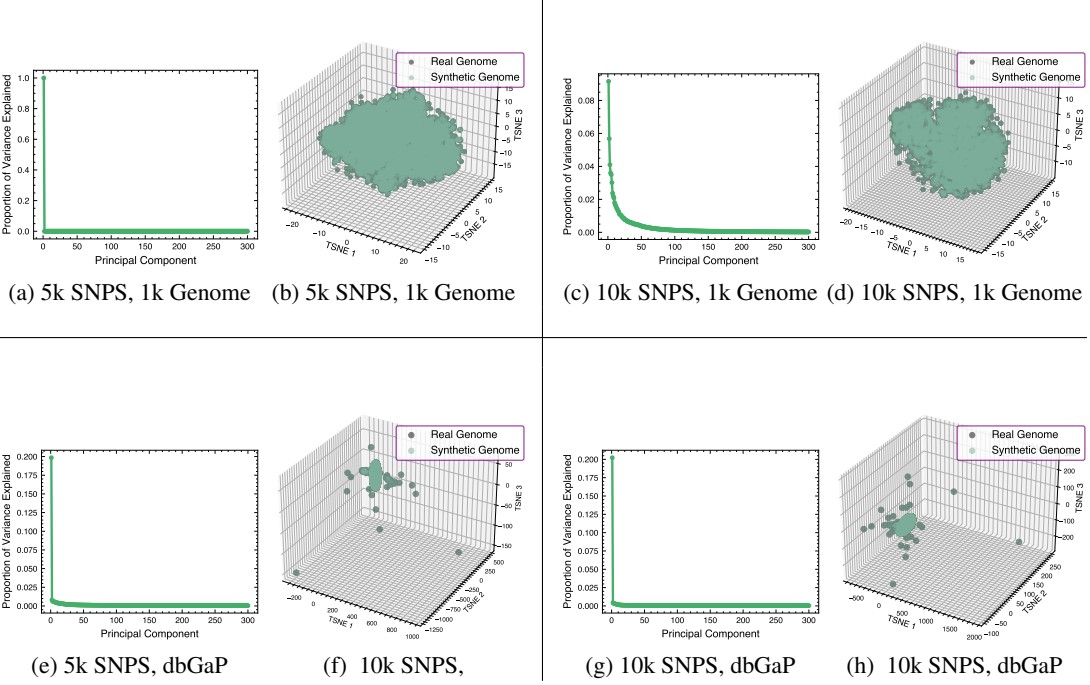

(a) 5k SNPS, 1k Genome    (b) 5k SNPS, 1k Genome    (c) 10k SNPS, 1k Genome   (d) 10k SNPS, 1k Genome

(e) 5k SNPS, dbGaP      (f) 10k SNPS,        (g) 10k SNPS, dbGaP     (h) 10k SNPS, dbGaP

Figure 7: The first row shows scree and TSNE plots for synthetic samples from Vanilla GAN trained on 1000 Genome data for 5k, and 10k SNPS respectively. The second row shows scree and TSNE plots for synthetic samples from WGAN trained on dbGaP data for 5k, and 10k SNPS respectively. The distance metric for the reconstruction loss is the hamming distance.

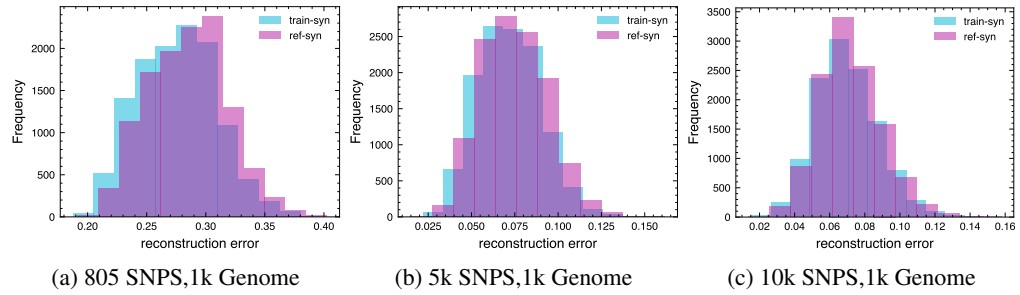

(a) 805 SNPS,1k Genome     (b) 5k SNPS,1k Genome     (c) 10k SNPS,1k Genome

Figure 8: The plots show the frequency distribution of reconstruction error for whole sequence memorization test for 3 batches of synthetic samples from vanilla GAN trained. The label **train-syn** (in seafoam green) indicates that we are measuring reconstruction loss for each batch of the synthetic samples with respect to the training data. For reference purposes, we also plot, **ref-syn**, which is the reconstruction loss of the synthetic samples given the reference samples (in purple). The distance metric for the reconstruction loss is the hamming distance.

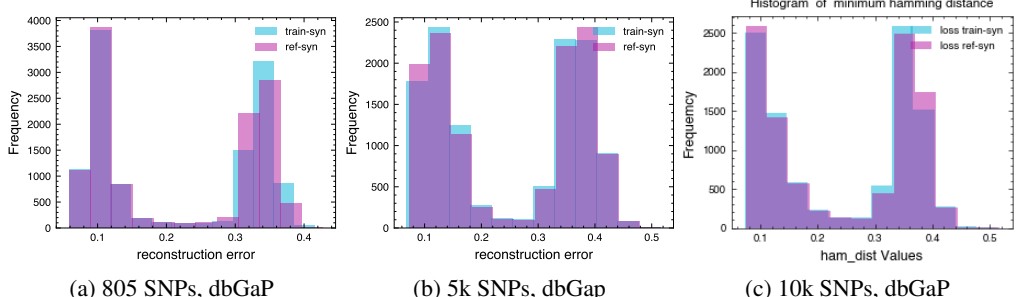

| (a) 805 SNPs, dbGaP | (b) 5k SNPs, dbGap | (c) 10k SNPs, dbGaP |

Figure 9: The plots show the frequency distribution of reconstruction error for 3 batches of synthetic samples from WGAN-GP. The label **train-syn** (in seafoam green) indicates that we are measuring reconstruction loss for each batch of the synthetic samples with respect to the training data . For reference purposes, we plot **ref-syn**, which is the reconstruction loss of the synthetic samples given the reference samples (in purple). The distance metric for the reconstruction loss is the hamming distance.

Table 4: Results for test to detect Overfitting

| Data Bank | GAN Variant | SNPS Dim. | MRE-gap | [§]p-value Train | p-value Ref. | [*]MMD Train | MMD Ref |
|---|---|---|---|---|---|---|---|
| 1000 genome | Vanilla GAN | 815 | 0.0045 | 0.0001 | 0.0001 | 0.0107 | 0.0137 |
| | | 5k | 0.0300 | 6.6e-05 | 6.6e-05 | 0.0094 | 0.0105 |
| | | 10k | 0.0108 | 5e-05 | 5e-05 | 0.0139 | 0.0151 |
| dbGaP | WGAN-GP | 815 | 0.0055 | 0.0001 | 0.0001 | 0.0194 | 0.0207 |
| | | 5k | 0.0143 | 6.6e-05 | 6.6e-05 | 0.1079 | 0.1106 |
| | | 10k | 0.002 | 5e-5 | 5e-5 | 0.0989 | 0.1018 |

[*] MMD = Maximum mean discrepancy. The values reported here are the unbiased estimate of $MMD^2$ Gretton et al. (2012b)
[§] p-values computed using kernel 2 sample test.

The normalized MRE-gap$_G$ is then defined as:

$$\text{MRE-gap}_G = \frac{MRE_G(\mathcal{X}_R)) - MRE_G(X_T)}{MRE_G(\mathcal{X}_R))}$$

**Two-kernel Test**     We carried out kernel 2 sample testGretton et al. (2012a) to quantify the similarity between the distribution of the synthetic samples and the distribution of the training data. The result of this test provides valuable insight regarding overfitting.

Specifically, let $X_T$ be of size $n$ and $X_S$ be synthetic samples from the GAN of size $N > n$. Then for each $x \in X_T$, let $x_s$ be the synthetic data sample satisfying the relation:

$$\arg\min_{x_s \in X_s} \delta(x, x_s)$$

where $\delta$ is the distance metric. Thus $x_s$ is the closest synthetic sample to the training data entry $x$. Let $\mathcal{S}^*$ contain the $x_s$ for each entry $x \in X_T$. Observe that $\mathcal{S}^*$ and $X_T$ are of the same size, $n$. We run the two kernel sample tests on the data samples $X_T$ and $\mathcal{S}^*$ as described in Gretton et al. (2012a).

In table 4, we present the normalized MRE-gap$_G$ for different target configurations. The figures seem to support that model seems not to be overfitting . Table 4 also reports the $p$-values and the unbiased estimate of the squared maximum discrepancy test from the kernel 2 samples test across different target configurations.

### F.3  DETECTOR ARCHITECTURE FOR GENOMIC DATA SETTING

Figure 10 is the Detector model architecture as implemented in Keras Chollet et al. (2015) and Tensorflow Abadi et al. (2015) for the MI attack against genomic data. The exact architecture differs depending on the size of the reference data data that is available to the adversary. The data size of dbGaP is larger than 1KG, thus the Detector architecture for all GANs trained on dbGaP has more layers than the corresponding architecture for target GAN trained on 1KG.

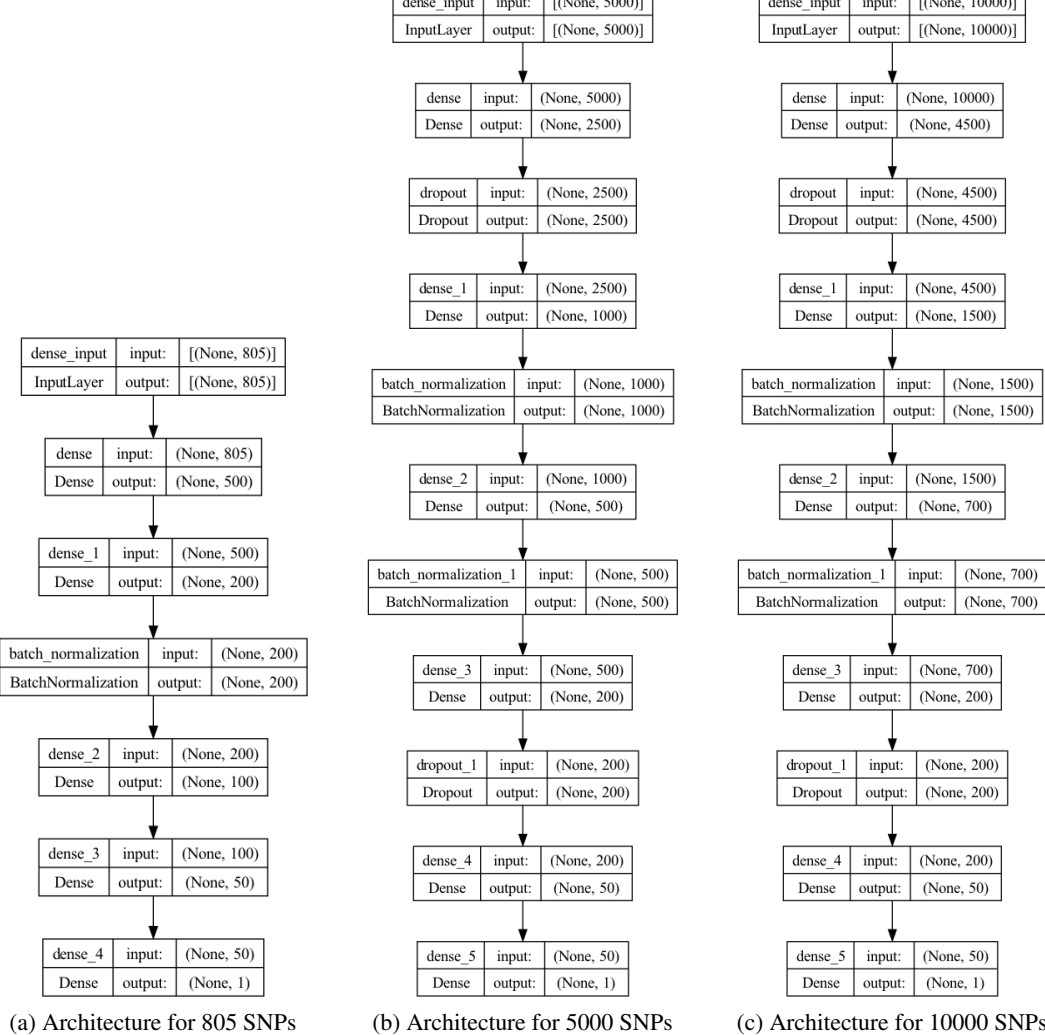

(a) Architecture for 805 SNPs  (b) Architecture for 5000 SNPs  (c) Architecture for 10000 SNPs

Figure 10: Detector Architecture for MIA against Vanilla GAN trained on 1000 genome database for 805,5000 and 10000 SNPS configurations. The Detector architecture varies depending on the SNPs configurations.

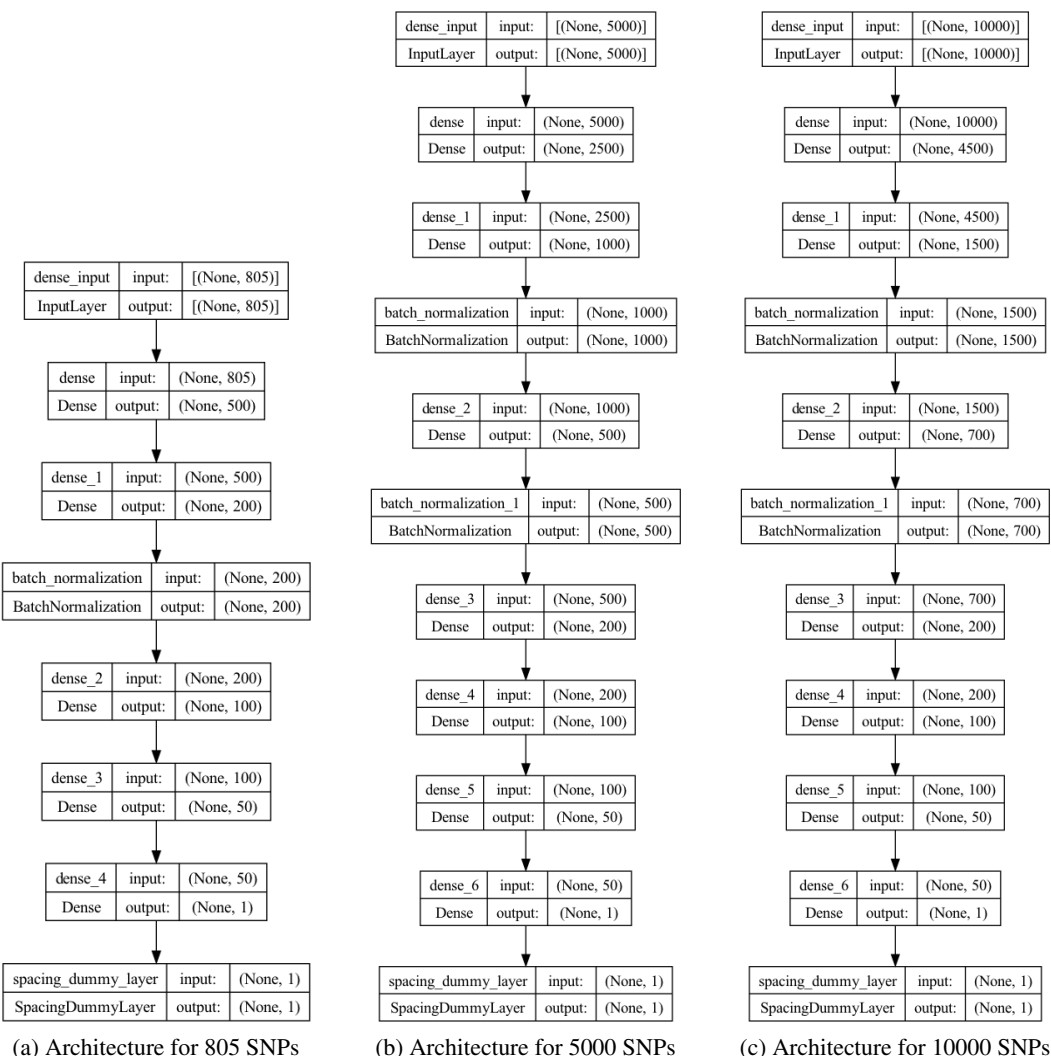

(a) Architecture for 805 SNPs   (b) Architecture for 5000 SNPs   (c) Architecture for 10000 SNPs

Figure 11: Detector Architecture for MIA against Vanilla GAN trained on 1000 genome database for 805,5000 and 10000 SNP dimension. The Detector architecture varies depending on the SNPs configurations.

Table 5: Target GAN Configurations for image data

| Data | Train Data Size | Ref. Data Size | Test Data Size | GAN Variant |
|------|-----------------|----------------|----------------|-------------|
| CIFAR10 | 40000 | 9000 | 2000 | BigGAN |
| | 40000 | 9000 | 2000 | Prog. GAN |
| | 40000 | 9000 | 2000 | DCGAN |
| | 4000 | 9000 | 2000 | Contra GAN |

# G  RESULTS FOR IMAGE GANS

**Distance Metric and Distance-based attack.**    The choice of metric for the distance-based attack on image GANs requires some careful consideration. Though L2 distance was suggested in  Chen et al. (2020), it should be noted that L2 distance suffers one major limitation when used as a metric for distance-based attack: it can't capture joint image statistics since it uses point-wise difference to measure image similarity  Fu* et al. (2023).

More recently, with the widespread adoption of deep neural networks, learning-based metrics have enjoyed wide popularity and acceptance  Dosovitskiy & Brox (2016); Gatys et al. (2015); Johnson et al. (2016). These metrics leverage pre-trained deep neural networks to extract features and use these features as the basis for metric computation  Fu* et al. (2023). LPIPs  Zhang et al. (2018) (Learned Perceptual Image Patch Similarity) is the most common of such learning-based metrics and was also proposed in  Chen et al. (2020) for carrying out distance-based attacks.

Our distance-based attack was conducted using both L2 distance and LPIPs. Both attack schemes were not very successful, though the LPIPS seemed to be slightly better than L2 for the distance attack.

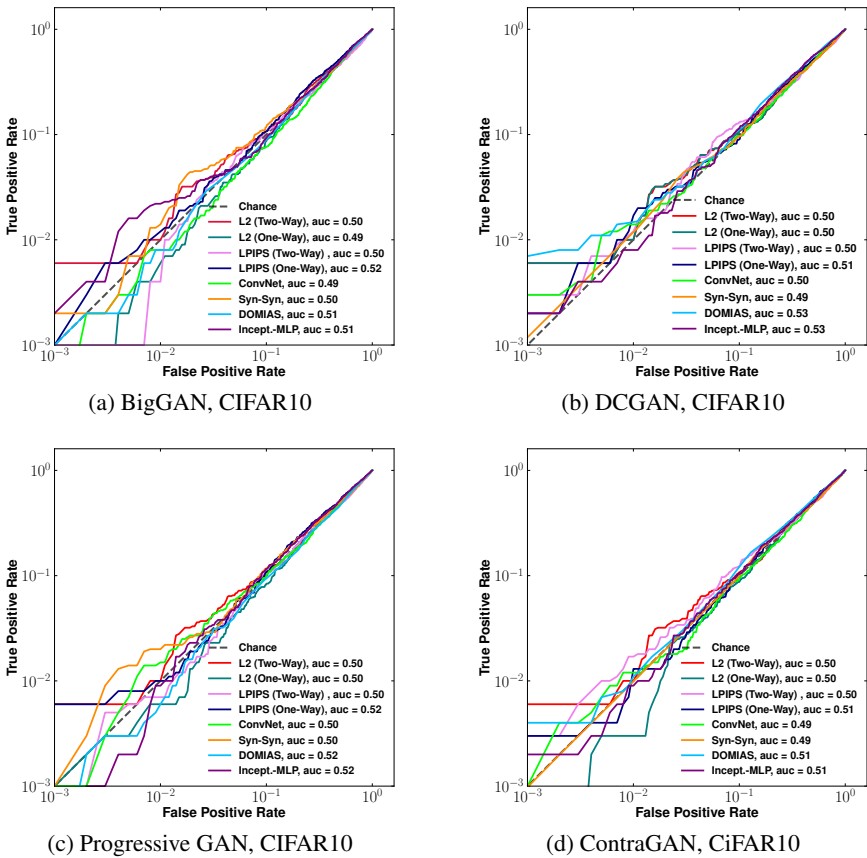

(a) BigGAN, CIFAR10        (b) DCGAN, CIFAR10

(c) Progressive GAN, CIFAR10        (d) ContraGAN, CiFAR10

Figure 12: The figures show the distance-based attack and the detector-based attacks against image GANs. Observe that the distance based attack are not particularly effective.

## H   DISTRIBUTION SHIFT EXPERIMENTS

Following prior work, we assume that our black-box attacks have access to fresh samples from the distribution $\mathcal{P}$. While this assumption is well-justified in many settings and both the Detector and distance-based attacks require access to reference samples, it is also reasonable to consider what happens to our attack success when the adversary does not have access to $\mathcal{P}$, but instead has access to a proxy distribution $\mathcal{P}_{\text{prox}}$. This setting is more realistic when the training samples used to train $G$ come from a highly sensitive distribution, for example patients with a specific rare disease, that the adversary may not have access to. We can then ask the question, how does the attack success of the Detector or distance-based attacks change as $d(\mathcal{P}, \mathcal{P}_{\text{prox}})$ grows? We consider two families of proxy distributions $\mathcal{P}_{\text{prox}}$ for dbGaP with 805 SNPs :

1. SNP-flipping: Let $p \in [0, 1]$. Then construct $\mathcal{P}_{\text{prox}}(p)$ by (i) sampling $x \sim \mathcal{P}$, (ii) selecting a random $p$ fraction of SNP positions in $x$ (note this is $805 * p$ positions. At each position $i$, let $\alpha_i$ be the mean allele frequency (MAF) at position $i$, which is the proportion of SNPS with a 1 in position $i$ as opposed to a 0. Re-sample a new value for position $i$ from Bernoulli($\alpha_i$). Note that when $p = 0$ we recover $\mathcal{P}_{\text{prox}}(0) = \mathcal{P}$, and when $p = 1$, we are sampling each SNP independently from its marginal distribution, and so increasing $p$ decreases the correlation between SNPs.

2. Mixture-distribution: We also create a family of proxy distributions that are a mixture between $\mathcal{P}$ on dbGaP, which we call $\mathcal{P}_{\text{1dbGaP}}$ and $\mathcal{P}_{\text{11KG}}$ where we've picked the same set of 805 SNPs across both datasets. Then $\mathcal{P}_{\text{prox}}(p) = p \cdot \mathcal{P}_{\text{1KG}} + (1 - p) \cdot \mathcal{P}_{\text{1dbGaP}}$. Again when

Table 6: Table of Attack Results (Image Data)

| GAN Variant | Dataset | Attack Method | TPR @0.01 FPR | TPR @0.1 FPR | TPR @0.005 FPR | TPR @0.001 FPR |
|---|---|---|---|---|---|---|
| BigGAN | CIFAR10 | two-way distance (L2) | 1.0% | 10.8% | 0.6% | 0.6% |
| BigGAN | CIFAR10 | one-way distance (L2) | 0.6% | 8.7% | 0.2% | 0.0% |
| BigGAN | CIFAR10 | two-way distance (LPIPS) | 0.4% | 11.0% | 0.1% | 0.1% |
| BigGAN | CIFAR10 | one-way distance (LPIPS) | 1.2% | 10.8% | 0.7% | 0.1% |
| BigGAN | CIFAR10 | Detector (Conv. net) | 0.8% | 7.7% | 0.3% | 0.0% |
| BigGAN | CIFAR10 | Detector (syn-syn) | 1.4% | 11.8% | 0.7% | 0.2% |
| BigGAN | CIFAR10 | Detector(Incept.-MLP) | 2.2% | 9.3% | 1.6% | 0.2% |
| DCGAN | CIFAR10 | two-way distance (L2) | 1.0% | 10.8% | 0.6% | 0.6% |
| DCGAN | CIFAR10 | one-way distance (L2) | 1.0% | 11.1% | 0.6% | 0.6% |
| DCGAN | CIFAR10 | two-way distance (LPIPS) | 1.2% | 13.0% | 0.7% | 0.3% |
| DCGAN | CIFAR10 | one-way distance (LPIPS) | 1.7% | 9.0% | 0.6% | 0.2% |
| DCGAN | CIFAR10 | Detector (Conv. net) | 1.4% | 9.7% | 1.1% | 0.3% |
| DCGAN | CIFAR10 | Detector (syn-syn) | 1.2% | 9.6% | 0.6% | 0.1% |
| DCGAN | CIFAR10 | Detector (Incept.-MLP) | 0.8% | 11.4% | 0.4% | 0.2% |
| ProjGAN | CIFAR10 | two-way distance (L2) | 1.0% | 10.7% | 0.6% | 0.6% |
| ProjGAN | CIFAR10 | one-way distance (L2) | 0.6% | 8.4% | 0.3% | 0.1% |
| ProjGAN | CIFAR10 | two-way distance (LPIPS) | 0.7% | 10.7% | 0.6% | 0.1% |
| ProjGAN | CIFAR10 | one-way distance (LPIPS) | 1.0% | 10.9% | 0.8% | 0.6% |
| ProjGAN | CIFAR10 | Detector (Conv. net) | 1.5% | 9.9% | 0.7% | 0.1% |
| ProjGAN | CIFAR10 | Detector (syn-syn) | 2.0% | 11.8% | 1.4% | 0.1% |
| ProjGAN | CIFAR10 | Detector (Incept.-MLP) | 0.9% | 11.6% | 0.2% | 0.0% |
| ContraGAN | CIFAR10 | two-way distance (L2) | 1.0% | 10.8% | 0.6% | 0.6% |
| ContraGAN | CIFAR10 | one-way distance (L2) | 0.3% | 8.7% | 0.3% | 0.0% |
| ContraGAN | CIFAR10 | two-way distance (LPIPS) | 1.7% | 12.3% | 1.0% | 0.3% |
| ContraGAN | CIFAR10 | one-way distance (LPIPS) | 1.3% | 10.4% | 0.4% | 0.3% |
| ContraGAN | CIFAR10 | Detector (Conv. net) | 1.2% | 9.6% | 0.7% | 0.1% |
| ContraGAN | CIFAR10 | Detector (syn-syn) | 0.9% | 10.0% | 0.5% | 0.01% |
| ContraGAN | CIFAR10 | Detector (Incept.-MLP) | 0.9% | 10.4% | 0.3% | 0.2% |

$p = 0$, $\mathcal{P}_{\text{prox}}(0) = \mathcal{P}$, and when $p = 1$ this corresponds to the adversary using solely 1KG data to orchestrate an attack on dbGaP.

Figure 13 plots $\log - \log$ ROC curves on both mixture distributions for the Detector attack. Figure 14 shows how AUC evolves for both mixture distributions as $p \to 1$, for the Detector attack and the two-way distance attack. We find that for both mixture distributions, as $p \to 1$, the attack success decreases (as measured by AUC and FPR at low TPR). We see that as we increase the proportion of flipped SNPs, thus decreasing the correlation between SNPs, attack success falls sharply before plateauing around AUC = .52. One explanation for this plateau is that beyond a certain point there is no information the Detector can extract from correlation between SNPs, but there is still relevant information present in the marginal distributions of the SNPs, which our SNP-flipping preserves. As we vary the mixture distribution proportion we again see a decrease in the attack AUC, but less marked than in SNP-flipping. This shows that 1KG is a reasonably good proxy for dbGaP at these SNPs, and highlights the practicality of the Detector as it can be executed without direct access to dbGaP data. For genomic data it is unsurprising that other datasets will be a very good proxy, as even when the underlying human populations are different, in most regions the genome will be very similar.

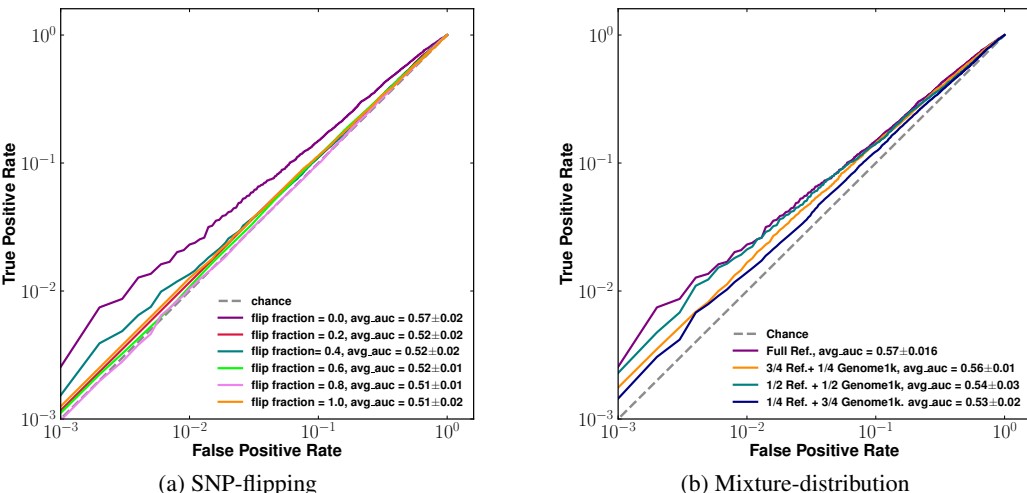

(a) SNP-flipping         (b) Mixture-distribution

Figure 13: SNP-flipping and Mixture-distribution ROC Curves for the Detector on Vanilla GAN trained on dbGaP 805 SNPs.

## I DIFFERENTIALLY PRIVATE GAN TRAINING

Differential privacy is the canonical way to provably prevent privacy attacks, see Section 2.4 of Zanella-Béguelin et al. (2022) for example, and there has been work on training differentially private GANs Xie et al. (2018); Jordon et al. (2018); Long et al. (2021); Tantipongpipat et al. (2019); Rosenblatt et al. (2020). We implement the PATECTGAN Jordon et al. (2018); Xu et al. (2019a) and DP-WGAN Xie et al. (2018), training on dbGaP with 805 SNPs. We set hyperparameters for private training of the DP-WGAN follows: learning rate$= 1e - 4$, batch size $= 256$, weight clip$=0.9$, epochs$=150$, latent vector ($z$) dimension $= 400$, $\epsilon = 1.72$, $\delta = 1e - 6$, noise multiplier$= 7$. For PATECGAN we set learning rate$= 1e - 4$, batch size $= 500$, epochs$=3$, latent vector ($z$) dimension $= 600$, $\epsilon = 3.$, noise multiplier $= 1e - 4$, samples per teacher $= 2000$, number of teachers $= 4$. We train on 805 SNPs because we have difficulty getting private GAN convergence at reasonable values of $\epsilon$ for higher dimensional datasets. In Figure 16 we plot the MIA attack success AUC for all our MIA attacks. We see that at $\epsilon = 1$ attack success essentially drops to the random baseline, but even at $\epsilon = 3$, attack AUC remains around .58. It is worth noting that even with this larger value of $\epsilon$, the TPRs at low FPRs for the PATECTGAN fail to beat random guessing, which is consistent with the theoretical guarantees from DP that bound the multiplicative ratio of TPR to FPR. We also include PCA plots showing how the private GAN has converged after training in Figure 15.

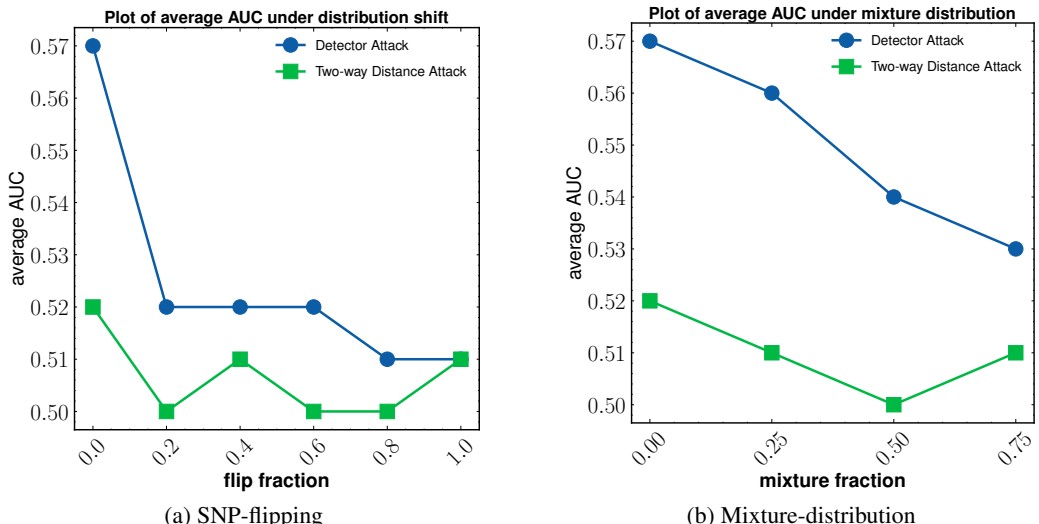

Figure 14: SNP-flipping and Mixture-distribution AUC vs mixture probability for the Detector and two-way distance attack on Vanilla GAN trained on dbGaP 805 SNPs.

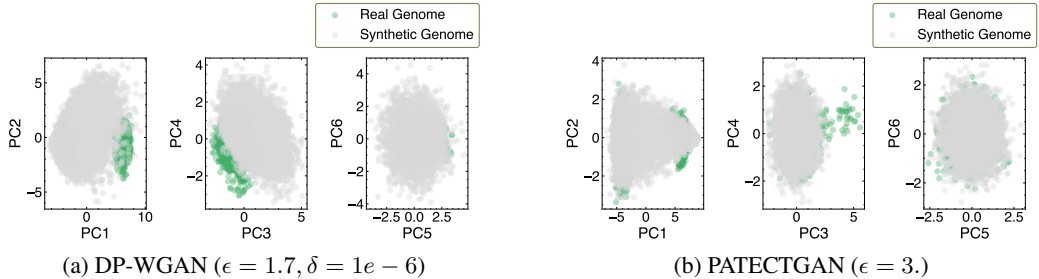

Figure 15: Convergence Plots for DP-GANS (DP-WGAN and PATECTGAN) trained on dbGaP, SNPs 805

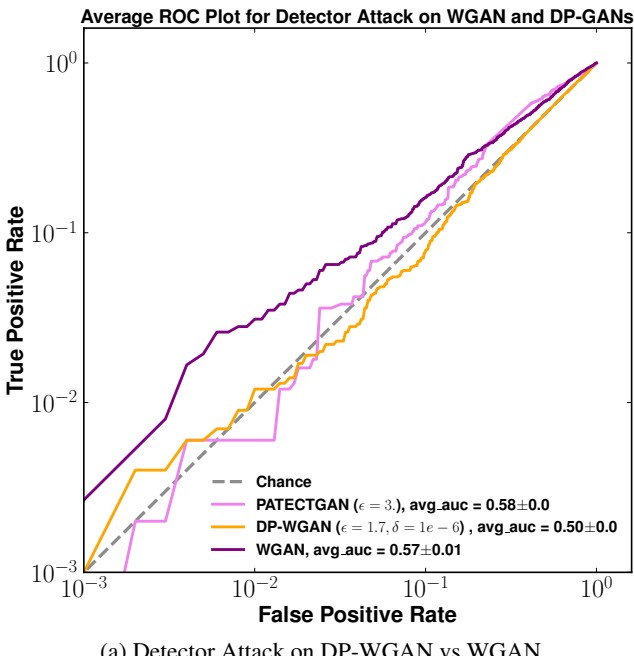

(a) Detector Attack on DP-WGAN vs WGAN

Figure 16: Detector attack on DP WGAN and WGAN

## J  ADDITIONAL RELATED WORK

Hilprecht et al. (2019) propose a black-box attack on GANs trained on CIFAR-10 that is very similar to a distance-based attack. Their statistic samples from the GAN and counts the proportion of generated samples that fall within a given distance $\epsilon$ of the candidate point. In order to set $\epsilon$, they estimate the $1\%$ or $.1\%$ quantiles of distances to the generated GAN via sampling. Note that this assumes access to a set of candidate points $x_i$ rather than a single candidate point that may or may not be a training point, although a similar idea could be implemented using reference data. While their attacks were quite effective against VAEs and on the simpler MNIST dataset, the best accuracy one of their distance-based attacks achieves on CIFAR-10 is a TPR that is barely above $50\%$, at a massive FPR that is also close to $50\%$ (Figure 4(c)) with their other distance-based method performing worse than random guessing. Relatedly, Chen et al. (2020) proposed a distance-based attack scheme based on minimum distance to a test sample. In Chen et al. (2020) the adversary has access to synthetic samples from the target GAN and synthetic samples from a GAN trained on reference data ($G_{ref}$). We follow this approach for distance-based attacks, which we discuss further in Section 4.2. The most closely related work to our detector methods is van Breugel et al. (2023b) who propose a density-based model called DOMIAS (Detecting Overfitting for Membership Inference Attacks against Synthetic Data), which infers membership by targeting local overfitting of the generative models. Rather than train a detector network to classify whether samples are generated from the target GAN or the reference data, DOMIAS performs dimension reduction in order to directly estimate both densities, and then uses the ratio of the densities as a statistic for membership inference.

Related to our study of tabular GANs trained on genomic data, Yelmen et al. (2021) proposed the use of GANs for the synthesis of realistic artificial genomes with the promise of none to little privacy loss. The absence of privacy loss in the proposed model was investigated by measuring the extent of overfitting using the nearest neighbor adversarial accuracy ($\text{AAT}_{TS}$) and privacy score metrics discussed in Yale et al. (2019). It should be noted that while over-fitting is sufficient to allow an adversary to perform MI (and hence constitute a privacy loss), it is not a necessary prerequisite for the attack to succeed, as shown in the formal analysis presented in Yeom et al. (2018).

