# OpenReview forum: "Black-Box Privacy Attacks Against GANs via Detector Networks"
_ICLR.cc/2024/Conference — Submitted to ICLR 2024_

### Official Review · Reviewer_rp3W · 2023-10-31

**Soundness:** 2 fair
**Presentation:** 2 fair
**Contribution:** 1 poor
**Rating:** 3
**Confidence:** 4

**Summary:**

The paper proposes black-box membership inference attacks against generative adversarial networks. It assumes that the attacker has access to the generated samples and the real distribution. The first attack is to train a classifier between the generated samples and the real samples and detect training samples if the predicted probability of being generated samples is larger than a threshold. The second attack detects training samples if the (relative) distance to generated samples is closer than a threshold. Experiments are conducted on an image dataset and a genomics dataset.

**Strengths:**

* Overall, the paper is well-written in terms of clarity.

**Weaknesses:**

* The proposed methods lack novelty. Some of the key methods have already been proposed in prior work published 3 years ago.

* The experiment results are not convincing enough.


Please see 'Questions' section for the details.

**Questions:**

The discussion of related work:
* The introduction claims that "Most existing work probing the privacy properties of GANs has typically focused on the "white-box" setting, where an adversary has access to the discriminator of the GAN, and is able to use standard techniques to conduct membership inference attacks based on the loss Chen et al. (2020); a notable exception is van Breugel et al. (2023a)." However, black-box attacks have already been proposed in LOGAN (Hayes et al. 2018) and GAN-leaks (Chen et al. 2020).

Novelty of the proposed approach:
The main approaches are almost the same as prior work published 3 years ago. In particular,
* The proposed detector-based attack is the same as case 1 of the discriminative setting in Section 4.4 of LOGAN (Hayes et al. 2018).
* The proposed one-way distance-based attack is the same as Section 5.2 of GAN-leaks (Chen et al. 2020)
* The proposed two-way distance-based attack is the same as Section 5.2 + Section 5.6 of GAN-leaks (Chen et al. 2020).
Please correct me in the rebuttal if I misunderstood it.

Concerns about the experiment results:
* Many plots in Figure 1 and Figure 2 show that the proposed approaches can perform worse than random guesses in some settings (e.g., Figure 1a, 1b, 1d, 1e, 1f, 1g, 1j, 1k). Do you have explanations for that?
* It claims that "... ADIS consistently outperforms the Detector ...". This is not the case in Figure 1j.
* Because many of the lines in Figure 1 and 2 have significant overlap and the rankings of the methods are not consistent across the entire range of FPR, it would be more convincing to conduct the experiments multiple times and report the average and std of the scores.

Other minor issues:
* Section 3: "Generative Adversarial Network(GANs)" should be "Generative Adversarial Network (GANs)"
* Section 3: "X_T ~ P" should be "X_T ~ P^n" where n is the number of training samples.
* Section 4.2: When expressing the generator reconstruction loss, the condition of R_L is mixed with G and X_G in different places. Please make the notation consistent.
* Section 4.2: remove the extra period in the section title.
* Figure 7a-7f: I doubt if we can conclude from these figures that the two point sets are close. Firstly, since only one viewpoint of the 3D plots is visualized,  it is impossible to know the exact location of each point. Moreover, in Figures 7b and 7d, most points from the "real genome" are not visible as they are covered by the points from the "synthetic genome".
* Section 5.1: "those results here ." should be "those results here."
* Section 6: BigGAn should be BigGAN.

---

> ### Author Response · Authors · 2023-11-22
> **Minor Comments (Part 1 of response to rp3W)**
>
> We thank the reviewer for their comments, we will address all minor comments and typos found, and update the related work to more accurately reflect LOGAN 2018, and GAN-leaks, both of which we do already cite. In the next comment we discuss the major points raised.
>
> **Minor Comments:**
>
> **Q1:**  Many plots in Figure 1 and Figure 2 show that the proposed approaches can perform worse than random guesses in some settings (e.g., Figure 1a, 1b, 1d, 1e, 1f, 1g, 1j, 1k). Do you have explanations for that?
>
> **R1:** On the genomic data, only the one-way distance attack really seems to perform worse than random guessing.
>
>  Recall that the one-way distance attack thresholds based on the distance between $x$ and the generated samples. We have added discussion of these results to Section 5.2:
> "Interestingly, the only method that consistently fails to beat the random baseline (AUC $< .5$) in all settings is the One-way distance-based attack. Paired with the moderate success of two-way distance attacks, this shows that for training points while the \emph{relative difference} between their distance to generated samples and their distance to test samples is larger than the difference in distances for test points, but the actual distance to the generated samples alone is not uniformly larger."
>
> **Q2:** It claims that "... ADIS consistently outperforms the Detector ...". This is not the case in Figure 1j.
>
> **R2**: We have updated the language to reflect that occasionally the detector outperforms ADIS.
>
> **Q3:** Because many of the lines in Figure 1 and 2 have significant overlap and the rankings of the methods are not consistent across the entire range of FPR, it would be more convincing to conduct the experiments multiple times and report the average and std of the scores.
>
> **R3:** All reported results are averaged over 11 training runs, and in Figure $1$ we report the AUC $\pm$ the standard deviation in the legend. We've added this to the caption of the figure to make this more clear.  We also added table $7$ to the Appendix that also contains the average AUCs and standard deviations.

---

> ### Author Response · Authors · 2023-11-22
> **Major Comments: Relationship to Prior Work (Part 2 of response to rp3W)**
>
> **Major Comments**:
>
> **Q1**: The proposed one-way distance-based attack is the same as Section 5.2 of GAN-leaks (Chen et al. 2020)
> The proposed two-way distance-based attack is the same as Section 5.2 + Section 5.6 of GAN-leaks (Chen et al. 2020). Please correct me in the rebuttal if I misunderstood it.
>
> **R1:** The distance-based attacks are similar to the attacks proposed in GAN-leaks, which we state in the contributions section:
> ``We compare our Detector-based methods to an array of distance and likelihood-based
> attacks proposed in prior work Chen et al. (2020).'' These distance based attacks are implemented to provide comparison to our other black-box attacks. We also note that a small contribution of this work is executing these distance-based attacks with different distance-metrics, like LPIPS (https://arxiv.org/abs/1801.03924). We also provide the most extensive evaluation of these distance-based attacks across tabular and image data.
>
> **Q2**: The proposed detector-based attack is the same as case 1 of the discriminative setting in Section 4.4 of LOGAN (Hayes et al. 2018).
>
> **R2:** We thank the reviewer for bringing this to our attention. We have updated the discussion in the draft to reflect this relationship to prior work. Although this diminishes the algorithmic novelty of our approach, we actually think that this **strengthens the impact of this particular paper.** There are two components to this, the algorithmic novelty, and the experimental finding.
>
> **Experimental Findings relative to LOGAN**: LOGAN studies training a detector on generated samples on CIFAR10 as we do in our Image GANs section and finds: “In Fig. 7, we plot the accuracy results for both settings, showing that the attack fails with both datasets when the attacker has only test set knowledge, performing no better than random guessing.” So the verdict on these black-box detector attacks from LOGAN is that they do not work. This stands in contrast to our findings, which is that in some settings, and particularly on genomic data, they do work very well. One reason is that since the LOGAN paper in 2018, our understanding of how to properly evaluate MIAs has evolved significantly since https://arxiv.org/abs/2112.03570 — rather than evaluating the accuracy of the MIA as in LOGAN, recent work including ours focusing on plotting the full ROC curve, and evaluating the TPR at low fixed FPR, which corresponds better to actual privacy risk for a small subgroup. By this metric on genomic data, the ADIS attack achieves a 10x improvement over the random baseline in some settings! On CIFAR10 our findings match LOGAN; although we do beat the random baseline at low FPRs for some detector-based methods, our improvements are modest, and worse than distance-based methods. **Since prior work concludes detector based attacks are not effective, the fact that our work shows they work very well on tabular data adds importance nuance.**
>
> **Novelty relative to LOGAN:** In addition to the important experimental findings above, relative to the LOGAN paper, we introduce the ADIS variant of the detector based attack, which performs much better in practice on genomic data. We also provide theoretical analysis of the detector success in Theorem 1 and in Equation 1, which may provide some insight on why the detector-based attacks perform worse in certain settings. Finally, we evaluate on new types of GAN architectures: BigGAN, Progressive GAN, and ContraGAN, and find differences in attack success across architectures.

---

> > ### Comment · Reviewer_rp3W · 2023-11-23
> > **Thank you!**
> >
> > I thank the authors for the detailed replies to all my questions!
> >
> > **Q3:** Sorry that I missed the statements that all reported results have already been averaged over 11 training runs. In that case, I would suggest the authors directly draw the standard error in all the figures (e.g., in shaded regions) so that it is easy to visualize the statistical significance of the results and the improvements.
> >
> > **Q1/Q2:**  For distance-based attacks, I know that the authors have mentioned the prior work. I also appreciate that the authors acknowledge that the only differences are in the distance metric (LPIPS) and the evaluation results. But in the current writing, the difference is not clearly highlighted. In fact, if I had not read Chen et al. (2020) before, purely based on the current writing, I might have misunderstood that these attacks are new.
> >
> > For the detector-based attack, I thank the authors for acknowledging its similarities to prior works. That being said, I agree with the authors that the paper still makes contributions, including new knowledge about these attacks. But again, the current paper presents it as a new approach, which is misleading and not true. If I were not familiar with LOGAN, I would not catch it too.
> >
> > In summary, the current writing makes it hard for readers to correctly judge the novelty of the work. Indeed, this already creates problems in the review process. For example, reviewer zbzo wrote "One of the main contributions of the paper is the proposition of two novel MIAs", but as we discussed above, both of them are *not* new.
> >
> > I would suggest the authors discuss the true contributions compared to the prior work more clearly in the revision. Such a change requires another round of review to give a fair judgment of the paper.

---

### Official Review · Reviewer_bnwi · 2023-11-01

**Soundness:** 2 fair
**Presentation:** 2 fair
**Contribution:** 3 good
**Rating:** 5
**Confidence:** 3

**Summary:**

This paper studies membership inference attacks against GAN in the black-box setting, where the adversary only has access to the generated synthetic data. The authors propose to leverage a detector trained to distinguish synthetic data from real data to predict membership. The efficacy of the proposed attack is empirically demonstrated on two genomic datasets and one image dataset.

**Strengths:**

- Membership inference attack against synthetic data is an underexplored but important research topic that has direct real-world applications, such as auditing private dataset release in the medical domain.

- Connecting the distinguishability between GAN-generated samples and real samples to the distinguishability between member samples and non-member samples is a good insight.

- The work evaluates privacy leakage by measuring TPRs at low FPRs, following Carlini et al., which is a very realistic setting.

- Besides standard image datasets, the experiments considered genomic datasets which have direct privacy implications.

**Weaknesses:**

- The attack assumes that the adversary can get fresh samples from the same distribution as the underlying private data. However, it is unclear whether such an assumption is reasonable in practice.

- Related to the above point, in real-world settings, oftentimes the adversary may only be able to get data samples from a slightly different distribution (e.g., to attack synthetic data released from one hospital, the adversary can only access data from another hospital). It is unclear how the proposed attack would perform under such distributional shifts. Some empirical results on this would be nice to have.

- The performance improvements of the proposed methods aren’t very significant compared to distance-based attacks and are not consistent across different settings. In many cases, distance-based attacks remain a very strong baseline (e.g., Fig1agij). In addition, the performance of the proposed attack seems to be very sensitive to the choice of architecture and dataset.

- The crux of the proposed attack is the connection between inferring G vs. P and inferring T vs. P, which is justified by Theorem 1. However, the proof is based on an oversimplified assumption that G is a linear mixture of P and T and thus does not generalize.

- The paper could benefit from a further round of proofreading to fix typos (e.g., that to random points -> than to, we trained average -> we average). Also, some notations in the Appendix are not consistent with the main paper (e.g., G = βP + (1 − β)T -> G = βT + (1 − β)D).

- In Theorem 1, it seems that beta should be in (0, 1) instead of [0, 1].

- In the proof of Lemma A.1, E_G[f(x)] should be 2TPR_M(f). Also, in the proof the Lipschitz constant is on the denominator which is not consistent with Eq. 1.

**Questions:**

1. How does the mode-collapse phenomenon of GAN motivate expressing G as a mixture of T and P?

2. How would the proposed attack perform if P is shifted from T?

---

> ### Author Response · Authors · 2023-11-22
> **New Experiments on Distribution Shift!**
>
> We thank the reviewer for their excellent suggestions. We have run a new battery of experiments to address their main critiques, which we feel has strengthened the paper. We first discuss the major comments, and then the minor ones.
>
> **Major Comments:**
>
> **Q1:**  How would the proposed attack perform if the distribution the adversary has access to is shifted away from the true distribution $\mathcal{P}$?
>
> **R1:** We note that although almost all prior work on MIAs assumes that adversary has access to the true distribution P, we agree that in some situations where data access is very limited for privacy reasons, it is reasonable to assume the adversary only has access to a proxy distribution. We’ve added new experiments evaluating how the attack success of our detector attack changes as the proxy distribution that the adversary has access to shifts away from $\mathcal{P}$. We find that under both models of distribution shift that we study, as the distribution shifts way from $\mathcal{P}$, attack success decreases. Surprisingly we find that attack success decreases less than we might have thought, showing that the detector is somewhat robust to distribution shift. We summarize theses results in the distribution shift section of the draft.
>
> **Q2:** Why is it reasonable to assume that $G$ is a convex combination of $P$ and $T$, $G = \alpha \mathcal{P} + (1-\alpha)T$.
>
> **R2:** First we remark that Equation 1 still holds for arbitrary $G$, which says we can bound the error of the detector attack by $W^1(G, T)$. Now Theorem $1$ attempts to sharpen this result by showing how even if $G$ is actually quite far from $T$, and $W^1(G, T)$ is quite large,  if $G$ is a convex combination of $\mathcal{P}, T$, the attack is still approximately optimal. We are not arguing that in practice $G$ is actually a convex combination of $\mathcal{P}, T$. We are more arguing that under this simplified representation of the generator, the attack is approximately optimal. While we do not expect that generators trained in practice will be an exact mixture distribution, the convex combination has some merits: (i) it reflects the fact that if properly trained G learns the true distribution $\mathcal{P}$ a $1-\alpha$ fraction of the time. Generative models have been observed to memorize their training data (e.g. https://arxiv.org/abs/2301.13188) and so the $\alpha$ component represents this memorization probability. Without a simplified model of the generator, obtaining theoretical results on attack optimality seems out of reach. We have updated the draft to provide this discussion in Section $4.1$.
>
> **Minor Comments:**
> Thank you for the found typos, we will make sure to fix all of the above prior to the camera ready, and we have fixed the typo with the lipschitz constant in the Appendix (it was correct in the main draft) thank you!

---

### Official Review · Reviewer_zbzo · 2023-11-01

**Soundness:** 4 excellent
**Presentation:** 3 good
**Contribution:** 4 excellent
**Rating:** 8
**Confidence:** 4

**Summary:**

This paper investigates the privacy protection provided by data synthesis methods based on generative adversarial networks (GANs). More precisely, new of set of membership inference attacks (MIAs) is proposed that only assume a black-box access to synthetic samples produced by the generative model. These attacks are compared to existing state-of-the-art approaches on a thorough set of experiments and the results obtained demonstrate that they outperform previous methods.

**Strengths:**

The introduction clearly reviews the state-of-the-art of MIAs against synthetic data and the challenges associated with such attacks.

One of the main contribution of the paper is the proposition of two novel MIAs, one based on training a detector to differentiate between training examples and examples coming from the distribution and the other based on the difference of distance between a particular sample to other synthetic samples as well as to reference samples. Both attacks are well motivated theoretically.

Experiments have been conducted on genomic data with two different GAN-based synthetic data generations as well as by comparing to a diverse set of existing state-of-the-art MIAs. The results obtained clearly demonstrate that the proposed attacks outperform previous MIAs. In addition, the results obtained with ADIS demonstrate that augmenting the feature space with distance-based characteristics increases significantly the success of the attacks.

Additional experiments were also conducted on image dataset with a wide range of GAN-based approaches.  The results obtained while significantly different than for genomic data also demonstrate the efficiency of the proposed attacks.

**Weaknesses:**

The writing of the paper is ok but could be improved (see for instance below for a few typos).

Overall, while the experiments conducted are very thorough, they should be complemented with at least one differentially-private data synthesis approach to assess in particular if such protection mechanism would impact the success of MIAs in a similar manner.

A few typos/remarks :
-« why detectors can be approximately optimal membership inference attacks » -> « why detectors are approximate optimal membership inference attacks
-The following reference is repeated twice : Boris van Breugel, Hao Sun, Zhaozhi Qian, and Mihaela van der Schaar. Membership inference attacks against synthetic data through overfitting detection, 2023b.
-« Rather than train » -> « Rather than training »
-« Absent any additional information » -> « In the absence of any additional information »
-« hamming » -> « Hamming »
-« requires a further choice of distance metric » -> « requires further a choice of a distance metric »
-« we only report those results here . » -> « we only report those results here. »
-« features improved attack success » -> « features mproves the attack success »
-The following sentence is not complete : « computation of reconstruction losses ?? »

**Questions:**

The main question that I have is with respect on how protection measure such as differential privacy would impact the conclusions drawn from the experiments. It would be good if the authors could do additional experiments to evaluate this issue.

---

> ### Author Response · Authors · 2023-11-21
> **Differentially Private GAN results added!**
>
> Thank you very much for your feedback, and the great suggestion about training a private GAN. We are incorporating all of the found typos to improve the readability of the paper, including unifying some notation between the draft and the Appendix, and adding more explanation of the Detector attack from the Appendix to the main draft.
>
> **Main Comment**: The main question that I have is with respect on how protection measure such as differential privacy would impact the conclusions drawn from the experiments. It would be good if the authors could do additional experiments to evaluate this issue.
>
> **Response**: In response to this suggestion, we trained two DP GANs, one as per https://arxiv.org/abs/1802.06739 which uses DP-SGD to privately update the discriminator. We track the privacy loss using the RDP accountant in tensorflow (https://github.com/tensorflow/privacy/blob/979748e09c416ea2d4f85e09b033aa9aa097ead2/tensorflow_privacy/privacy/analysis/compute_dp_sgd_privacy.py). We also trained a DP GAN using PATE-GAN (https://openreview.net/forum?id=S1zk9iRqF7) although this did not work as well as the DP-SGD GAN, since the sample splitting required resulted in either too few samples to train the target GANs, or too much noise when privately aggregating. As a result we report results for the DP-SGD GAN, at epsilon = 1.7 in the new DP GAN training section of the draft.
>
> On 805 dbGaP the Detector attack achieves AUC of .57 against the non-private WGAN. Against the DP-WGAN trained using DP-SGD, the attack AUC is only .50, which is the same as random guessing. We note that the fact training with DP destroys MIA attack success isn't that surprising, as differential privacy provides theoretical defense guarantees against $\textit{any MIA}.$ See Theorem 3.1 in https://arxiv.org/pdf/2010.12112.pdf which says that if the training algorithm is $(\epsilon, \delta)$-DP, then for any attack the $ \text{TPR} \leq \text{FPR}+ (e^\epsilon - 1 + 2\delta)/(e^\epsilon + 1)$. The difficulty in use DP as a defense is that historically training private classifiers that achieve competitive accuracy has been challenging, let alone DP generative models. While this result is encouraging, we note that it is possible a more complete battery of GAN convergence/utility tests may find that our private GAN is less useful than the non-private GAN even at $805$ SNPs, and we anticipate difficulty scaling these results to higher dimensions. We leave this as an intriguing direction for future work!

---

### Official Review · Reviewer_QeZK · 2023-11-03

**Soundness:** 3 good
**Presentation:** 1 poor
**Contribution:** 2 fair
**Rating:** 5
**Confidence:** 5

**Summary:**

The authors present several novel black box membership inference attacks against GANs. These attacks broadly fall under two buckets: i) Detector based (classifier based), ii) Distance based. The authors evaluate these attacks on several datasets and compare them with each other as well as comparable algorithms from other papers.

**Strengths:**

1) The problem of black-box membership inference attacks against GANs is an interesting one. It is also probably the most practical and understudied form of membership inference attacks against GANs.
2) The algorithms presented in the paper seem novel and advance the state of the art.
3) The empirical evaluations of the algorithms are very reasonable and I particularly liked the choice of metrics.

**Weaknesses:**

1) The authors do not cite several relevant papers which have looked at black box membership inference attacks e.g. [1,2]
2) Please include a more detailed explanation of the ADIS attack in the main paper since it is a novel contribution of this paper and has been used in the experiments.
3) Figure 3 is cited in the main paper but is not a part of the main paper. Please at mention clearly that the figure is a part of the appendix and try to write the paper such that the appendix is purely optional reading.
4) In theorem 4.1, why is it reasonable to assume that G is a convex combination of P and T? Please clarify/explain in the paper.
5) In section 5.1, explain how the different train/test set sizes were chosen. Seems a little arbitrary.



[1] Mukherjee, Sumit, et al. "privGAN: Protecting GANs from membership inference attacks at low cost to utility." Proc. Priv. Enhancing Technol. 2021.3 (2021): 142-163.
[2] Xu, Yixi, et al. "Mace: A flexible framework for membership privacy estimation in generative models." arXiv preprint arXiv:2009.05683 (2020).

**Questions:**

See weaknesses.

---

> ### Author Response · Authors · 2023-11-21
> **Simple Model of the Generator as a Mixture**
>
> Thank you very much for your comments.
>
> **Minor Comments:** We will add the relevant citations on black-box MIAs, move the description of ADIS up to the main draft, and added a description of how the detector pipeline works, so that Figure 4 is not essential to read the paper, as it is in the Appendix. Train and test sizes for the detector attack were chosen so that there are enough training points to train (i) a target GAN that converges (ii) a Detector that is accurate on a test set and (iii) points left over to accurately evaluate the MIA attack success. Typically we need a few thousand points each to train the generative models and the detector, and then at least $500$ points to evaluate MIA attack success.
>
> **Major Comments**:
>
> **Q:** Why is it reasonable to assume that $G$ is a convex combination of $P$ and $T$, $G = \alpha P + (1-\alpha)T$.
>
> **R:** First we remark that Equation 1 still holds for arbitrary $G$, which says we can bound the error of the detector attack by $W^1(G, T)$. Now Theorem $1$ attempts to sharpen this result by showing how even if $G$ is actually quite far from $T$, and $W^1(G, T)$ is quite large,  if $G$ is a convex combination of $P, T$, the attack is still approximately optimal. We are not arguing that in practice $G$ is actually a convex combination of $P$, $T$. We are more arguing that under this simplified representation of the generator, the attack is approximately optimal. While we do not expect that generators trained in practice will be an exact mixture distribution, the convex combination has some merits: (i) it reflects the fact that if properly trained G learns the true distribution P a $1-\alpha$ fraction of the time. Generative models have been observed to memorize their training data (e.g. https://arxiv.org/abs/2301.13188) and so the $\alpha$ component represents this memorization probability. Without a simplified model of the generator, obtaining theoretical results on attack optimality seems out of reach. We have updated the draft to provide this discussion in Section $4.1$.
>
> We have also made additional improvements to the paper during the rebuttal period, in response to the other reviewers, including:
> (i) an investigation of attack success when the adversary does not have access to reference samples, and instead must use samples from a shifted distribution
> (ii) the impact of training a DP GAN on MIA attack success

---

### Author Response · Authors · 2023-11-21
**New Draft Uploaded**

We have uploaded a new rebuttal draft where we have implemented all requested changes, except for minor typos and a few references which can be easily fixed for the camera ready. We thank the reviewers for the amazing feedback which has resulted in $2$ new sections of the paper, one focusing on DP GAN training as a potential defense against MIAs, and the other investigating how attack success changes with distribution shift. These sections are currently in the Appendix, but we will consider moving them to the main draft if the paper is accepted.

---

### Meta-Review · Area_Chair_R79b · 2023-12-08

**Metareview:**

This paper proposes several new black-box membership inference attacks against GANs. For both distance-based and detector-based attacks, the authors introduce new techniques such as alternative distance functions and ADIS detector that improve the effectiveness of resulting MIAs.

Reviewers raised several weaknesses, including lack of comparison against some baseline black-box MIA methods (Reviewer QeZK), and that empirical performance does not improve consistently over prior SOTA. The more critical weakness is regarding how the proposed techniques are positioned within the existing literature. Given that both distance-based and detector-based attacks have been previously introduced, based on the paper's writing, it is unclear which part of the proposed attacks are novel. This also relates to the weakness of empirical performance improvement over SOTA since it's unclear which components are the most effective. The authors are strongly encouraged to improve the paper's writing to better distinguish themselves from prior work, and perform careful ablation studies to demonstrate the value of their proposed techniques.

**Justification For Why Not Higher Score:**

Unclear writing and positioning within the existing literature makes it difficult to identify the authors' contributions. Empirical evaluation is also inconclusive.

**Justification For Why Not Lower Score:**

N/A

---

### Decision · Program_Chairs · 2024-01-16

Reject